# The zinc-finger transcription factor Hindsight regulates ovulation competency of *Drosophila* follicles

Lylah D Deady[1], Wei Li[1], Jianjun Sun[1,2]*

[1]Department of Physiology and Neurobiology, University of Connecticut, Connecticut, United States; [2]Institute for Systems Genomics, University of Connecticut, Connecticut, United States

**Abstract** Follicle rupture, the final step in ovulation, utilizes conserved molecular mechanisms including matrix metalloproteinases (Mmps), steroid signaling, and adrenergic signaling. It is still unknown how follicles become competent for follicle rupture/ovulation. Here, we identify a zinc-finger transcription factor Hindsight (Hnt) as the first transcription factor regulating follicle's competency for ovulation in *Drosophila*. Hnt is not expressed in immature stage-13 follicle cells but is upregulated in mature stage-14 follicle cells, which is essential for follicle rupture/ovulation. Hnt upregulates Mmp2 expression in posterior follicle cells (essential for the breakdown of the follicle wall) and Oamb expression in all follicle cells (the receptor for receiving adrenergic signaling and inducing Mmp2 activation). Hnt's role in regulating Mmp2 and Oamb can be replaced by its human homolog Ras-responsive element-binding protein 1 (RREB-1). Our data suggest that Hnt/RREB-1 plays conserved role in regulating follicle maturation and competency for ovulation.

DOI: https://doi.org/10.7554/eLife.29887.001

*For correspondence:
jianjun.sun@uconn.edu

Competing interests: The authors declare that no competing interests exist.

## Introduction

Ovulation is a complex process of releasing fertilizable oocytes from mature follicles and is essential for animal reproduction (*Espey and Richards, 2006*). To ensure successful ovulation, a follicle must be developed to full maturity to be competent to receive an ovulatory stimulus and to activate proteolytic systems for follicle rupture. Several proteolytic systems have been found to regulate follicle rupture in vertebrates, including matrix metalloproteinase (Mmp), plasminogen activator/plasmin, and ADAMS-TS (*Curry and Smith, 2006*; *Takahashi et al., 2013*). In addition, a surge of luteinizing hormone (LH) serves as a master regulator to initiate the ovulation event and activates the EGF/EGFR-Ras-MAPK signaling pathway to propagate the ovulatory signal from outer granulosa cells to inner cumulus cells in the preovulatory follicles (*Conti et al., 2012*; *Fan et al., 2009*, *2011*, *2012*; *Hsieh et al., 2007*). However, molecular mechanisms coupling the Ras-MAPK pathway to the activation of proteolytic systems for follicle rupture are largely unknown.

Ovulation in *Drosophila* utilizes conserved molecular mechanisms and involves a follicle rupture process to release mature oocytes from the ovary. *Drosophila* have two ovaries, connected at their posterior ends by bilateral oviducts (*Figure 1*). Each ovary contains ~16 ovarioles, where egg chambers are assembled in the germarium at the anterior and develop through 14 characteristic stages toward the posterior end (*Spradling, 1993*). Each egg chamber contains one oocyte and 15 nurse cells surrounded by a layer of somatic follicle cells. In stage-14 egg chambers (also named mature follicles), all nurse cells are degraded, leaving an oocyte surrounded by follicle cells; Matrix metalloproteinase 2 (Mmp2) is upregulated in posterior follicle cells (*Figure 1*; *Deady et al., 2015*). In addition, *Oamb (octopamine receptor in mushroom body)*, encoding an α-adrenergic receptor-like G-protein-coupled receptor for octopamine (OA), is also upregulated in all follicle cells of stage-14

**eLife digest** The release of an egg from the ovary of a female animal is a process known as ovulation. Animals as different as humans and fruit flies ovulate in largely similar ways. Yet the systems involved in controlling ovulation are still not well understood. An egg cell develops within a collection of cells that help the egg to form properly. Together, this unit is called a follicle. During ovulation, connections between the egg and the rest of the follicle break down and the egg is eventually ejected.

Ovulation happens in response to a hormone signal from the brain. In humans, this hormone is called luteinizing hormone, whereas in flies it is called octopamine. Specialized protein molecules on the surface of the follicle cells receive these hormone signals, but can only cause ovulation in mature follicles. It was not clear what allows only mature follicles to ovulate.

Deady et al. have now used the fruit fly *Drosophila melanogaster* to examine ovulation to identify how the process is controlled. The results showed that a protein called Hindsight primes follicle cells for ovulation. When a follicle reaches its final stage (called stage 14 in flies), the gene for Hindsight becomes active and produces the protein. This protein then activates other genes. One of the activated genes makes a protein that receives the hormone signal, while another makes a protein that breaks down follicle cells and allows the egg to be released.

The findings of Deady et al. reveal that Hindsight is needed for ovulation in flies. Further experiments then showed that the gene for equivalent human protein can be transplanted into flies and can still prime follicles for ovulation. This indicates that the genes in humans and flies may perform the same tasks. Studying ovulation is an important part of understanding female fertility and could help scientists to understand more about human reproduction. These results may also lead to new contraceptives and improved approaches for treating infertility.

DOI: https://doi.org/10.7554/eLife.29887.002

egg chambers (*Lee et al., 2003*; *Deady and Sun, 2015*). OA, released from terminal nerves that innervate ovaries, activates Oamb receptor in stage-14 follicle cells, which induces calcium rise and activates Mmp2 (*Deady and Sun, 2015*; *Heifetz et al., 2014*; *Middleton et al., 2006*; *Monastirioti, 2003*). Mmp2 enzymatic activity leads to degradation of posterior follicle cells and release of the encapsulated oocyte (called follicle rupture; *Figure 1*; *Deady et al., 2015*). The rest of the follicle cells remain at the end of the ovariole to form a corpus luteum (*Deady et al., 2015*). Local adrenergic signaling has also been suggested to regulate mammalian ovulation but no molecular mechanisms have been illustrated (*Kannisto et al., 1985*; *Schmidt et al., 1985*). In parallel to progesterone signaling in mammalian ovulation, ecdysteroid signaling is also activated in stage-14 follicle cells and is essential for *Drosophila* ovulation; ecdysteroid signaling modulates OA/Oamb-induced Mmp2 activation, but does not affect Oamb expression nor Mmp2 expression (*Knapp and Sun, 2017*). Thus, it is currently unknown what induces Mmp2 and Oamb expression in stage-14 follicle cells and how these follicles become fully competent for ovulation.

The zinc-finger transcription factor Hindsight (Hnt; encoded by gene *pebbled*) contains 14 C2H2 zinc-finger domains and is homologous to mammalian Ras-responsive element-binding protein 1 (RREB-1). Both Hnt and RREB-1 bind to similar DNA sequences, and human RREB-1 can functionally replace Hnt in attenuating expression of *nervy* and *hnt* itself in *Drosophila* salivary gland (*Ming et al., 2013*). RREB-1 functions downstream of the Ras-MAPK pathway to either suppress or promote Ras target genes in multiple tissues including colon, thyroid, and pancreatic cancers (*Kent et al., 2010*, *2013*; *Mukhopadhyay et al., 2007*; *Thiagalingam et al., 1996*; *Zhang et al., 2003*). Hnt is expressed in a variety of tissues in development and plays multiple developmental roles including control of embryonic germ band retraction (*Yip et al., 1997*; *Reed et al., 2001*), regulation of retinal cell fate and morphogenesis (*Pickup et al., 2002*; *Wilk et al., 2004*; *Pickup et al., 2009*; *Oliva and Sierralta, 2010*; *Oliva et al., 2015*), maintenance of tracheal epithelial integrity (*Wilk et al., 2000*, *2004*), and differentiation of spermathecae and intestinal stem cells (*Sun and Spradling, 2013*; *Baechler et al., 2015*). Hnt is also expressed in follicle cells of stage 7-10A egg chambers, where it functions as a downstream target of Notch signaling to suppress Hedgehog signaling and to induce the mitotic/endocycle transition (*Sun and Deng, 2007*). Hnt continues its

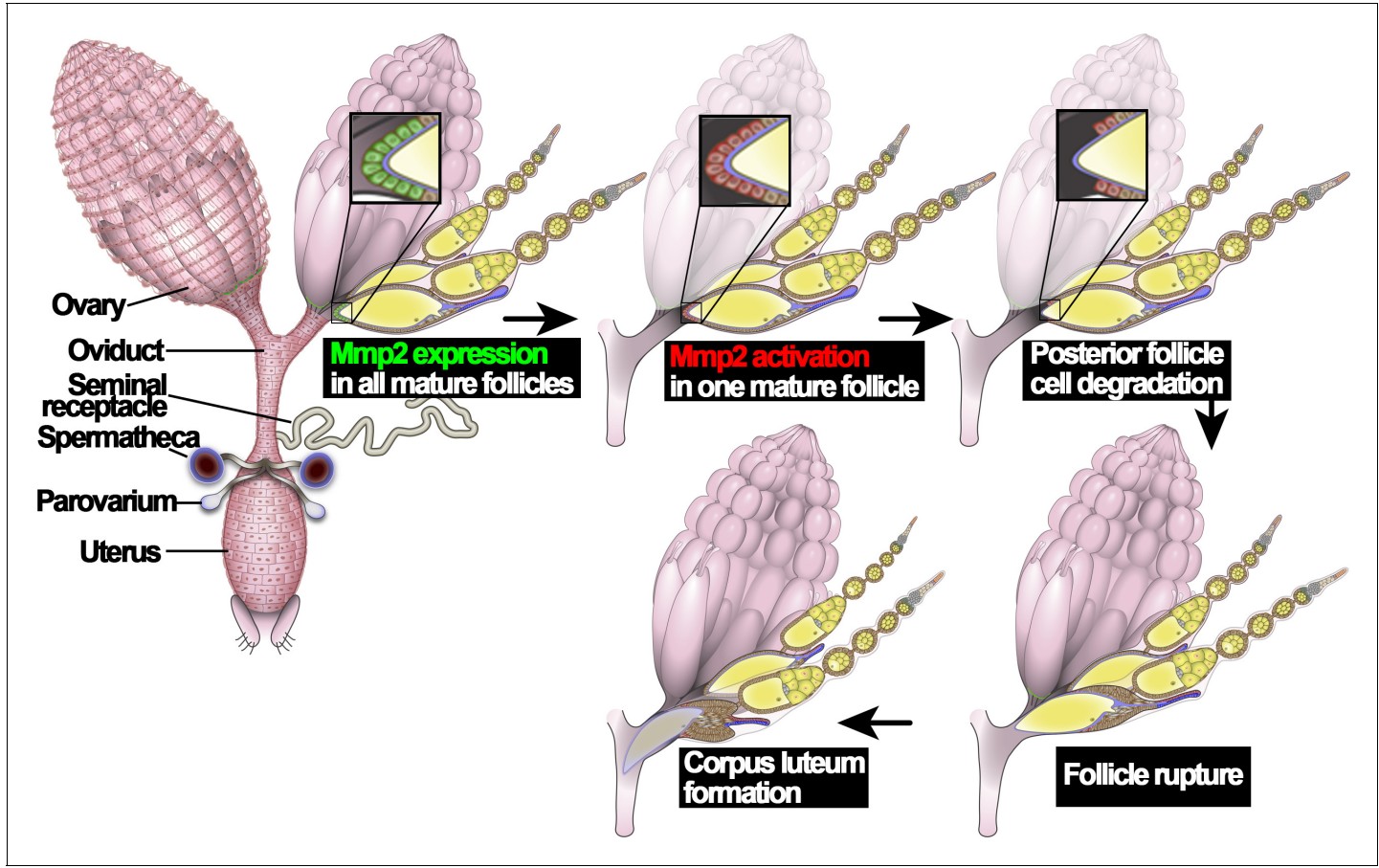

**Figure 1.** An illustration of *Drosophila* ovulation process. The female reproductive system, consisting of two ovaries, oviduct, uterus, seminal receptacle, and a pair of spermathecae and parovaria, was depicted in the cartoon. Two representative ovarioles with different staged egg chambers were highlighted in the right ovary. Oocytes and nurse cells are in yellow. Mmp2 expression is shown in green and Mmp2 activity is shown in red.

DOI: https://doi.org/10.7554/eLife.29887.003

expression in anterior follicle cells throughout late oogenesis. In contrast, Hnt expression in main-body follicle cells is downregulated from stage 10B to stage 13 and re-upregulated in stage-14 (*Deady et al., 2015*), where its role is unknown. Moreover, few downstream targets of Hnt have been identified and its relationship to Ras signaling is also unknown.

Here, we characterized the dynamic expression of Hnt in stage-14 follicle cells. By using molecular and genetic tools, we demonstrated that Hnt expression in stage-14 follicle cells is essential for follicle rupture partly by upregulation of Oamb and Mmp2 expression in these follicle cells. Thus, Hnt functions as an essential transcription factor to prime follicles to be competent for follicle rupture/ovulation. In addition, Hnt's role in follicle rupture can be replaced by human RREB-1. Our data, along with the involvement of Ras-MAPK signaling in mammalian ovulation, lead us to propose that Hnt/RREB-1 has a conserved role in regulating follicle rupture/ovulation downstream of Ras-MAPK signaling pathway.

## Results

### Dynamic expression of Hindsight in stage-14 follicle cells

Hnt is not expressed in stage-13 follicle cells except those at the anterior region; however, it is upregulated in all stage-14 follicle cells and the corpus luteum (*Deady et al., 2015*). Upon closer examination, we found three distinct patterns of Hnt expression throughout stage-14 egg chambers: (I) high Hnt expression in anterior and posterior but low/no Hnt in the middle follicle cells ('A/P-Hnt'

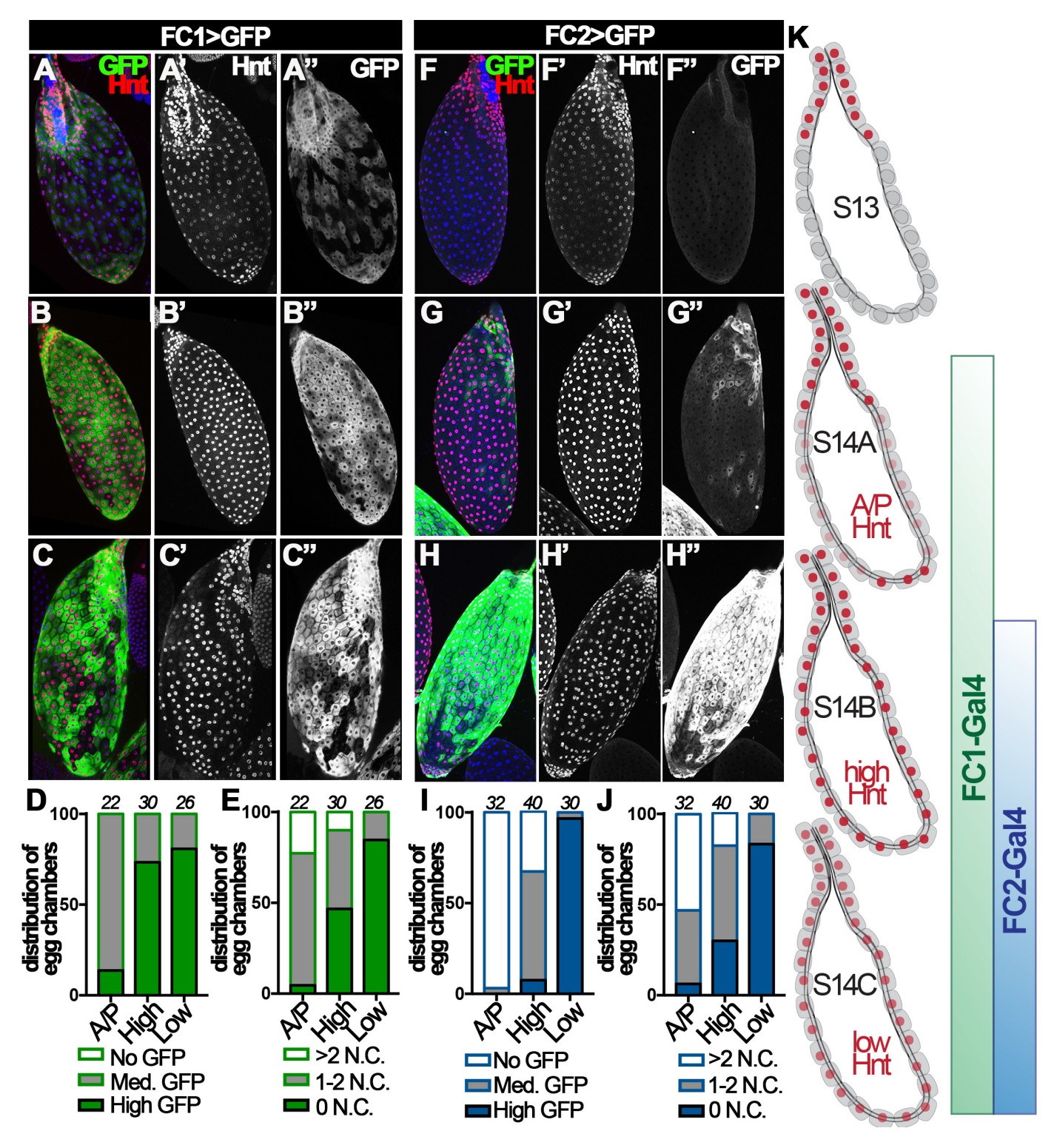

**Figure 2.** Hindsight expression in stage-14 egg chambers. (A–C) Hnt expression (red in A-C and white in A'-C') in A/P-Hnt (A), high-Hnt (B), and low-Hnt (C) egg chambers. FC1 expression (FC1 Gal4 driving *UAS-eGFP*, *FC1 > GFP*) is shown in green (A–C) and white in (A"–C"). Nuclei are shown by DAPI in blue (A–C). (D–E) Quantification of FC1 expression (D) and residual nurse cell nuclei (E) in A/P-Hnt, high-Hnt, and low-Hnt egg chambers. The number of stage-14 egg chambers analyzed is noted above each bar. (F–H) Hnt expression (red in F-H and white in F'-H') in A/P-Hnt (F), high-Hnt (G), and low-Hnt (H) egg chambers. FC2 expression (*FC2 > GFP*) is shown in green (F–H) and white (F"–H"). Nuclei are shown in blue (F–H). (I–J) Quantification of FC2 expression (I) and residual nurse cell nuclei (J) in A/P-Hnt, high-Hnt, and low-Hnt egg chambers. (K) A schematic cartoon shows

*Figure 2 continued on next page*

*Figure 2 continued*
the temporal pattern of Hnt, FC1 and FC2 expression in stage-14 egg chambers. FC1-related graphs are colored green and FC2-relaed graphs are colored blue.
DOI: https://doi.org/10.7554/eLife.29887.004
The following figure supplement is available for figure 2:

**Figure supplement 1.** Egg chambers with different patterns of Hindsight expression.
DOI: https://doi.org/10.7554/eLife.29887.005

egg chambers; *Figure 2A and F*, *Figure 2—figure supplement 1A–B*); (II) high Hnt expression in all follicle cells ('high-Hnt' egg chambers; *Figure 2B and G*); (III) low Hnt expression in all follicle cells ('low-Hnt' egg chambers; *Figure 2C and H*; also see *Figure 2—figure supplement 1C–D*). To determine the developmental sequence of the aforementioned three types of stage-14 egg chambers, we analyzed the expression patterns of Hnt against the expression of a stage-14 follicle-cell Gal4 driver (44E10-Gal4; renamed as FC1 for simplicity) (*Deady and Sun, 2015*) and the number of nurse cell nuclei in these egg chambers. 86% of A/P-Hnt egg chambers had medium-level GFP expression driven by FC1, while more than 73% of high-Hnt and 80% of low-Hnt egg chambers had high-level GFP expression (*Figure 2A–D*). This indicates that A/P-Hnt egg chambers are the youngest, which is consistent with the observation that A/P-Hnt egg chambers typically have more residual nurse-cell nuclei than the other two types of egg chambers (*Figure 2E*). In addition, high-Hnt egg chambers still contained one or two nurse-cell nuclei, while low-Hnt egg chambers typically did not contain nurse-cell nuclei and were skinner and dehydrated (*Figure 2E* and *Figure 2—figure supplement 1D*). Because nurse-cell nuclei are progressively degraded starting around stage-12 by a non-cell-autonomous mechanism to generate fully matured egg chambers, which have no nurse cell nuclei and are dehydrated (*Drummond-Barbosa and Spradling, 2004*; *Timmons et al., 2016*), the above analysis demonstrates that high-Hnt egg chambers are at the intermediate stage, while low-Hnt egg chambers are the most mature egg chambers.

This conclusion was further supported by additional analysis using a late stage-14 follicle-cell Gal4 driver (47A04-Gal4; renamed as FC2 for simplicity) (*Deady and Sun, 2015*). Consistent with the previous result, both A/P- and high-Hnt egg chambers had no or minimal GFP expression driven by FC2, while low-Hnt egg chambers had highest GFP expression and fewest nurse-cell nuclei (*Figure 2F–J*). Altogether, these analyses demonstrate that Hnt is first upregulated in posterior follicle cells, filled in across the entire egg chamber, and then overall downregulated in follicle cells of fully matured egg chambers (*Figure 2K*). Therefore, we propose to categorize stage-14 egg chambers into three distinct stages and rename A/P-, high-, and low-Hnt egg chambers as stage-14A, stage-14B, and stage-14C egg chambers, respectively (*Figure 2K*).

## Hindsight in stage-14 follicle cells is required for normal ovulation

The dynamic Hnt expression in stage-14 follicle cells prompted us to investigate its function in follicle maturation and ovulation. To bypass the early requirement of Hnt in follicle cell differentiation (*Sun and Deng, 2007*), we used RNA interference (RNAi) to deplete Hnt expression specifically in stage-14 follicle cells with FC1 or FC2 Gal4 driving *UAS-hnt$^{RNAi}$* expression. While FC1 started to be expressed in stage-14A follicle cells, it was weak to deplete Hnt expression in stage-14A follicle cells (*Figure 3—figure supplement 1A–D*), but became progressively more efficient in older follicle cells with two independent *hnt$^{RNAi}$* lines (*Figure 3A–D* and *Figure 3—figure supplement 1E–H*); more than 80% of stage-14C egg chambers had no detectable Hnt expression in their follicle cells. In contrast, FC2 started to be expressed in stage-14B follicle cells and most effectively depleted Hnt expression in stage-14C follicle cells except with *hnt$^{RNAi2}$*, which only showed strong reduction in ~43% of egg chambers (*Figure 3E–H* and *Figure 3—figure supplement 1I–P*).

Females with RNAi-mediated *hnt* depletion in stage-14 follicle cells (named *hnt$^{RNAi}$* females for simplicity) were then assayed for egg-laying ability. *hnt$^{RNAi}$* females laid significantly fewer eggs than control females after mating (*Figure 3I*). This phenotype was manifested by using both stage-14 follicle-cell Gal4 drivers and with two independent *hnt$^{RNAi}$* lines, which demonstrates that Hnt expression in stage-14 follicle cells was essential for normal egg laying. The decrease in egg-laying number was not caused by an oogenesis defect, as plenty of stage-14 egg chambers were observed before

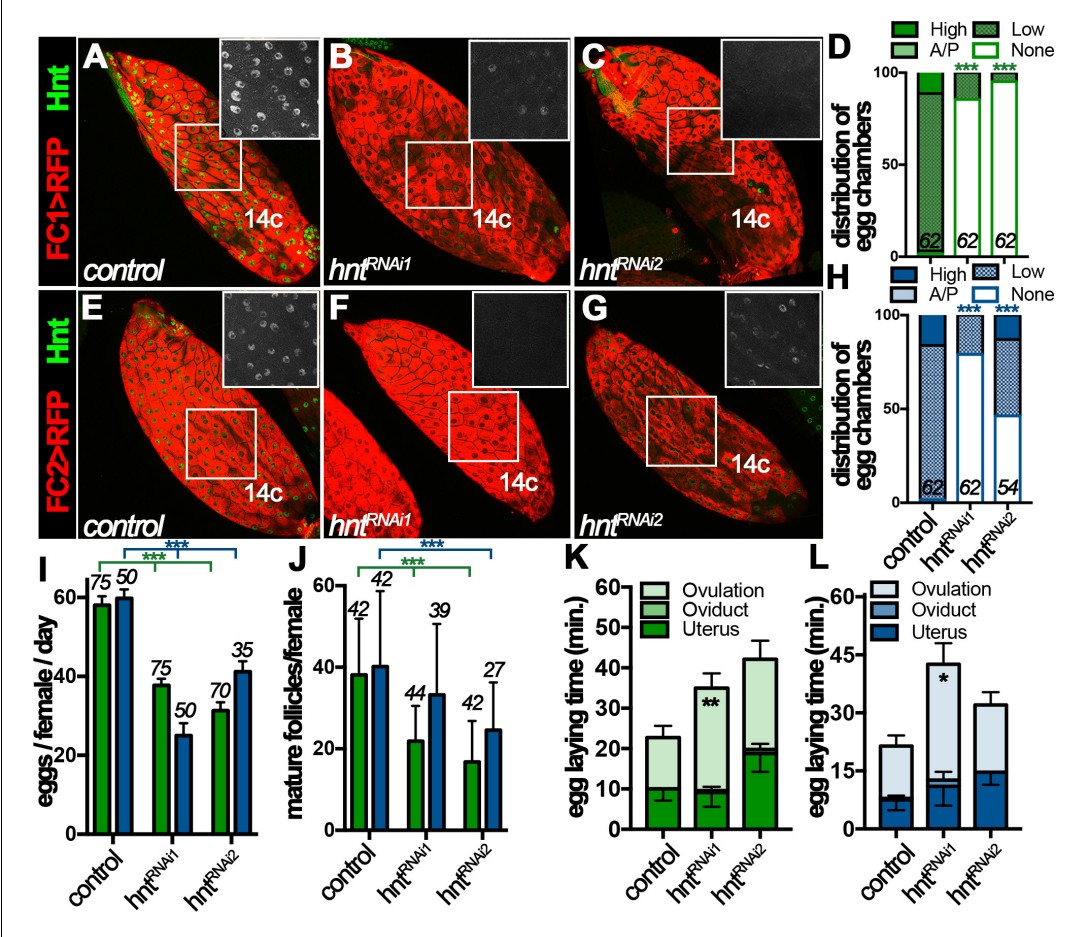

**Figure 3.** Hindsight expression in stage-14 follicle cells is essential for normal ovulation. (A–D) Hnt expression (green) in control (A) and *hnt*$^{RNAi}$ (B–C) stage-14C egg chambers with FC1. FC1 expression (*FC1 > RFP*) is shown in red. Inserts are high magnification of Hnt expression in squared areas. The quantification of Hnt expression (categorized as high-Hnt, low-Hnt, A/P-Hnt, and None-Hnt) in stage-14C egg chambers is shown in (D). The number of stage-14C egg chambers (selected according to no nurse-cell nuclei/high FC1 expression) analyzed is noted above each bar. (E–H) Hnt expression (green) in control (E) and *hnt*$^{RNAi}$ (F–G) stage-14C egg chambers with FC2. FC2 expression (*FC2 > RFP*) is shown in red. Inserts are high magnification of Hnt expression in squared areas. The quantification of Hnt expression in stage-14C egg chambers is shown in (H). The stage-14C egg chambers are selected according to no nurse-cell nuclei/high FC2 expression. (I–J) The quantification of egg laying (I) and mature egg chambers in each female's ovaries after egg laying (J) in control or *hnt*$^{RNAi}$ females with FC1 (green bars) or FC2 Gal4 (blue bars). The number of females is noted above each bar. (K–L) The egg-laying time in control or *hnt*$^{RNAi}$ females with FC1 (K) or FC2 (L). Also see **Supplementary file 1**. *p<0.05, **p<0.01, ***p<0.001.

DOI: https://doi.org/10.7554/eLife.29887.006

The following figure supplements are available for figure 3:

**Figure supplement 1.** Hnt expression in stage-14A and stage-14B egg chambers.
DOI: https://doi.org/10.7554/eLife.29887.007

**Figure supplement 2.** *hnt* depletion in stage-14 follicle cells does not affect oogenesis in virgin females.
DOI: https://doi.org/10.7554/eLife.29887.008

and after egg-laying experiments (**Figure 3J** and **Figure 3—figure supplement 2**). The egg-laying process consists of ovulation (the release of egg from the ovary into the oviduct), egg transportation through the oviduct, and oviposition (the release of egg in the uterus to the outside environment). To determine which step in the egg-laying process was affected in *hnt*$^{RNAi}$ females, we utilized our previously developed method to estimate the average time required for each step in the egg-laying process (**Sun and Spradling, 2013**; **Deady and Sun, 2015**; **Knapp and Sun, 2017**). Consistent with our previous data, control females spent 12–14 min to ovulate an egg, less than a minute to transport egg through the oviduct, and eight - 10 min to hold an egg in the uterus and oviposit (**Figure 3K–L** and **Supplementary file 1**). In contrast, *hnt*$^{RNAi}$ females spent more than 25 min to

ovulate an egg, which was significantly longer than the control females (*Figure 3K–L* and *Supplementary file 1*). These data demonstrate that Hnt in stage-14 follicle cells is required for normal ovulation.

## Hindsight in stage-14 follicle cells is necessary for OA-induced follicle rupture

Ovulation consists of a breakdown of posterior follicle cells and a subsequent rupture of oocyte into the lateral oviduct (*Figure 1*), which is induced by octopaminergic signaling and can be recapitulated in an *ex vivo* culture system (*Deady and Sun, 2015*). The requirement of Hnt for normal ovulation led us to hypothesize that Hnt is required for OA-induced follicle rupture. Consistent with this idea, about 45% stage-14 egg chambers isolated according to FC1 expression from control females ruptured in response to OA stimulation, whereas fewer than 10% of egg chambers from $hnt^{RNAi}$ females ruptured in response to OA stimulation (*Figure 4A–D*). In addition, more than 85% stage-14 egg chambers isolated according to FC2 expression from control females ruptured in response to OA stimulation (*Figure 4E and H*), consistent with the fact that FC2 is expressed in more mature egg chambers than FC1 (*Figure 2*). In contrast, egg chambers isolated from $hnt^{RNAi1}$ or $hnt^{RNAi2}$ females with FC2 ruptured at the rate of 10% and 33%, respectively (*Figure 4F–H*). Consistent with this result, follicles isolated from *hnt* transheterozygous mutant females also showed significant reduction in OA-induced follicle rupture in comparison to control follicles (*Figure 4—figure supplement 1*). These results demonstrate that Hnt is required in stage-14 follicle cells for follicle rupture.

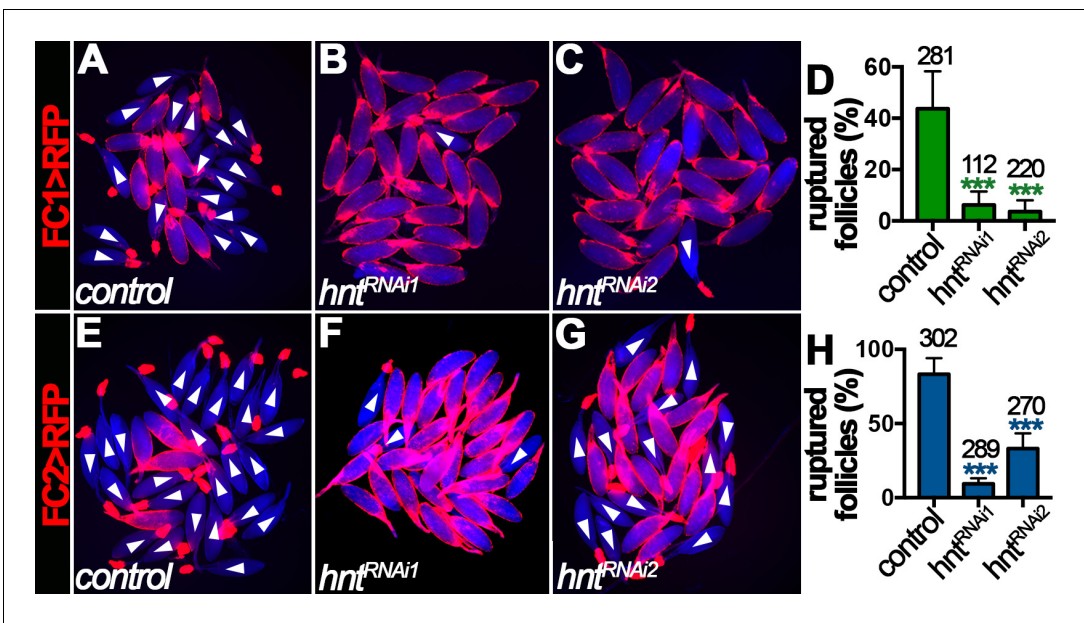

**Figure 4.** Hindsight is required for OA-induced follicle rupture. (**A–D**) Representative images show control (**A**) and $hnt^{RNAi}$ (**B–C**) egg chambers with FC1 after three-hour culture with 20 µM OA. Quantification of OA-induced follicle rupture is shown in (**D**). (**E–H**) Representative images show control (**E**) and $hnt^{RNAi}$ (**F–G**) egg chambers with FC2 after three-hour culture with 20 µM OA. Quantification is shown in (**H**). Egg chambers were isolated according to *FC1 > RFP* (red in A-C) or *FC2 > RFP* (red in E-G) expression. Bright-field images of the egg chambers are shown in blue, and ruptured egg chambers are marked by white arrowheads. The number of egg chambers is listed above each bar. ***p<0.001.

DOI: https://doi.org/10.7554/eLife.29887.009

The following figure supplement is available for figure 4:

**Figure supplement 1.** *hnt* mutant mature follicles are defective for OA-induced follicle rupture.

DOI: https://doi.org/10.7554/eLife.29887.010

## Hindsight is essential for Mmp2 activity in posterior follicle cells of stage-14 egg chambers

Mmp2 is essential for follicle rupture, and its enzymatic activity is activated by OA stimulation (*Deady and Sun, 2015*). To determine whether Hnt regulates follicle rupture by influencing Mmp2 activity, we assayed Mmp2 enzymatic activity in egg chambers from control and $hnt^{RNAi}$ females after OA stimulation *ex vivo*. After a three-hour incubation with OA, ~60% of control egg chambers isolated according to FC1 expression had posterior gelatinase activity (*Figure 5A and D*), whereas only ~25% of $hnt^{RNAi}$ egg chambers had posterior gelatinase activity (*Figure 5B–D*). In addition, about 90% of control egg chambers isolated according to FC2 had posterior gelatinase activity, in contrast to 25% and 47% of $hnt^{RNAi1}$ and $hnt^{RNAi2}$ egg chambers, respectively (*Figure 5E–H*). The proportion of follicles with posterior gelatinase activity was correlated to the proportion of follicles that ruptured, and both were significantly decreased in $hnt^{RNAi}$ egg chambers, which strongly supports that Hnt controls follicle rupture by regulating Mmp2 activity.

To further support this notion and to avoid the possibility that the above observed phenomenon is an artifact of *ex vivo* culture, we determined whether Hnt indeed regulates Mmp2 activity *in vivo*. One of the known substrates of Mmp2 is the basement-membrane (BM) protein collagen IV, encoded by *viking* (*vkg*), during imaginal disc morphogenesis and fatbody dissociation at pupal development (*Srivastava et al., 2007*; *Jia et al., 2014*). Vkg is detected in the basement membrane of follicle cells throughout oogenesis and we reasoned it could be a substrate of follicular Mmp2 as well. We found that 70% of FC2-expressing egg chambers had lost follicular Vkg protein in a large posterior area (open-BM configuration), 26% had lost Vkg protein in a small posterior area (broken-BM configuration), and 4% had intact Vkg protein (intact-BM configuration) at their posterior end (*Figure 5I–J and O*). When *tissue inhibitor of matrix metalloproteinase* (*Timp*, encoding an inhibitor of Mmp enzymatic activity) was overexpressed in stage-14 follicle cells using FC2, the BM configuration was dramatically shifted toward intact-BM configuration (*Figure 5K and O*), indicating that Mmp activity is responsible for the degradation of Vkg at the posterior end of stage-14 egg chambers. In addition, RNAi-mediated *Mmp2* depletion in stage-14 follicle cells showed a similar trend as overexpression of *Timp*, although less effectively (*Figure 5L and O*), demonstrating that Mmp2 is, at least partially, responsible for the Vkg degradation. Furthermore, *hnt* depletion in stage-14 follicle cells also shifted BM configuration toward broken- and intact-BM configuration as *Mmp2* depletion (*Figure 5M–O*). Altogether, these data demonstrate that Hnt regulates Mmp2 activity, which is responsible for Vkg degradation at the posterior end of stage-14 egg chambers during ovulation.

## Hindsight is required for Mmp2 expression in posterior follicle cells at stage 14

OA binds to Oamb receptor in stage-14 follicle cells, which leads to a rise of intracellular $Ca^{2+}$ concentration and subsequent activation of Mmp2 (*Deady and Sun, 2015*). To elucidate the mechanism of Hnt in regulating Mmp2 activity, we sought to determine whether Hnt interferes with OA/Oamb-$Ca^{2+}$-Mmp2 pathway upstream and/or downstream of $Ca^{2+}$ rise. In comparison to OA, $Ca^{2+}$ ionophore ionomycin was sufficient to induce more than 95% control egg chambers to rupture at the end of a three-hour culture, regardless whether egg chambers were isolated according to FC1 or FC2 expression (*Figure 6A,D,E and H*). In contrast, ionomycin was still not sufficient to induce follicle rupture in $hnt^{RNAi}$ egg chambers (except those with FC2 driving $hnt^{RNAi2}$ expression; *Figure 6B–D and F–H*), despite that it was able to induce $Ca^{2+}$ rise in follicle cells (*Figure 6—figure supplement 1A–D*, *Videos 1–3*). The incompetency of $hnt^{RNAi}$ egg chambers to ionomycin stimulation indicates that Hnt regulates components downstream of $Ca^{2+}$ rise in the OA/Oamb-$Ca^{2+}$-Mmp2 pathway; the almost normal response to ionomycin but defective response to OA in $hnt^{RNAi2}$ egg chambers with FC2 indicates that Hnt also regulates components upstream of $Ca^{2+}$ rise in the OA/Oamb-$Ca^{2+}$-Mmp2 pathway. Consistent with this idea, OA was not sufficient to induce $Ca^{2+}$ rise in $hnt^{RNAi}$ egg chambers with FC2 (*Figure 6—figure supplement 1E–H*, *Videos 4–6*). Since Hnt is first upregulated in posterior follicle cells (*Figure 2*), where Mmp2 is expressed (*Deady et al., 2015*), and then swept across the entire follicle cells (*Figure 2*), where Oamb is expressed (*Lee et al., 2003*; *Deady and Sun, 2015*), we hypothesize that Hnt regulates both Mmp2 and Oamb expression in stage-14 follicle cells.

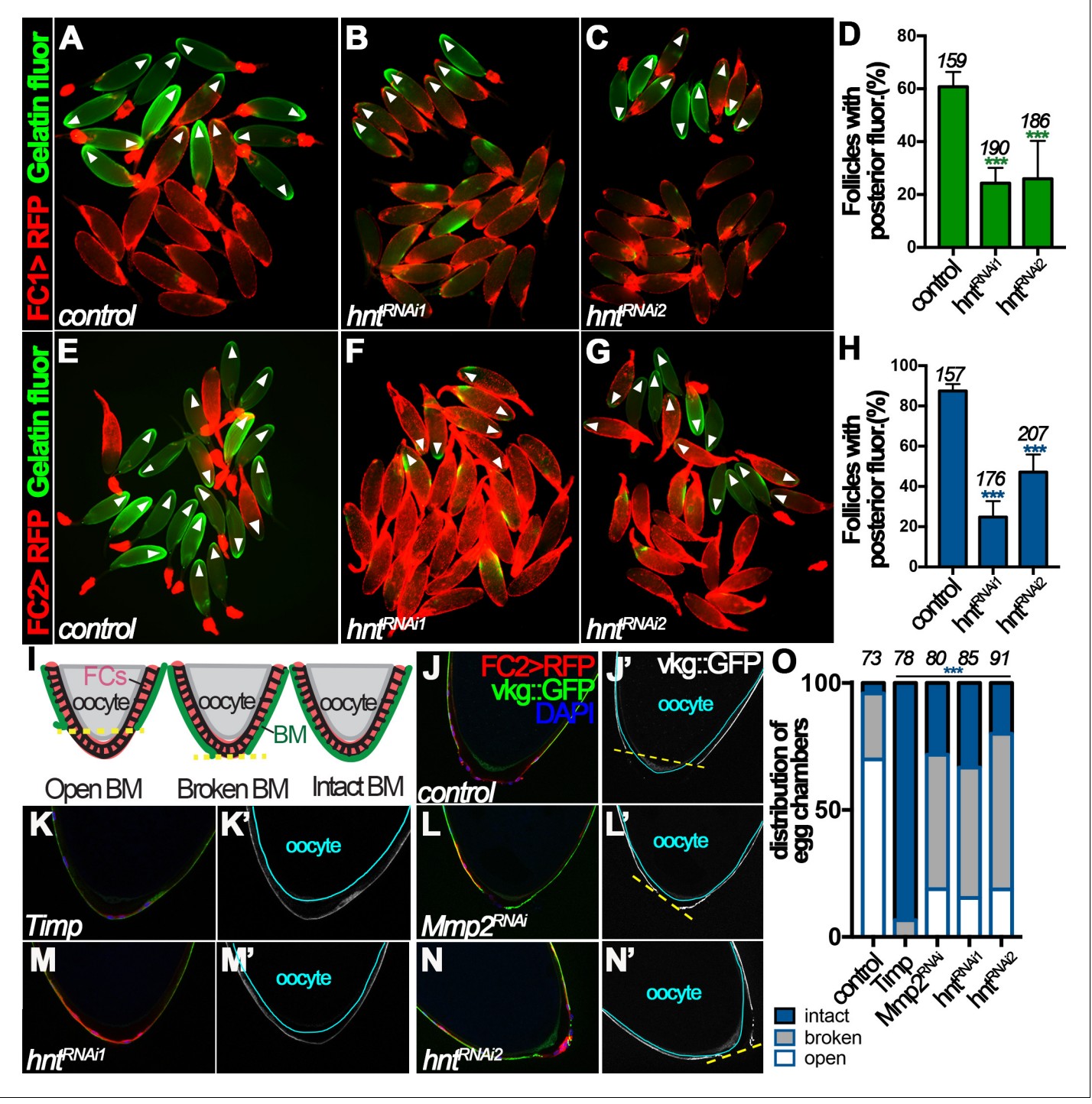

**Figure 5.** Hindsight regulates Mmp2 activity in stage-14 follicle cells. (A–C) Representative images show OA-induced Mmp2 activity (green) in control (A) and $hnt^{RNAi}$ (B–C) egg chambers with FC1 after three-hour culture with OA. Egg chambers with posterior Mmp2 activity are marked by arrowheads. Quantification is shown in (D). (E–H) Representative images show OA-induced Mmp2 activity (green) in control (E) and $hnt^{RNAi}$ (F–G) egg chambers with FC2 after three-hour culture with OA. The quantification is shown in (H). (I) A diagram shows the three categories of basement-membrane (BM) configurations (according to Vkg::GFP expression) of follicle cells in isolated stage-14 egg chambers. When a line connecting the posterior-most Vkg edges bisects the oocyte, it is defined as an open-BM configuration, whereas when the line does not bisect the oocyte, it is defined as a broken-BM configuration. The intact-BM configuration is defined as intact, continuous Vkg::GFP throughout the posterior of the egg chamber. (J–J') A control egg chamber identified according to FC2 >RFP expression shows the open-BM configuration. (K–K') An egg chamber with overexpression of *Timp* shows the intact-BM configuration. (L–L') An $Mmp2^{RNAi}$ egg chamber shows the broken-BM configuration. (M–N) $hnt^{RNAi}$ egg chambers show intact-BM (M–
*Figure 5 continued on next page*

Figure 5 continued

M') and broken-BM (N–N') configurations. (O) Quantification of BM configurations of FC2-expressing egg chambers with respective genotypes. The number of egg chambers analyzed is noted above each bar. ***p<0.001.

DOI: https://doi.org/10.7554/eLife.29887.011

To investigate the role of Hnt in Mmp2 expression, we examined Mmp2 expression using a *Mmp2::GFP* fusion gene in the endogenous locus. Consistent with our previous report, Mmp2::GFP was detected in posterior follicle cells of stage-14 egg chambers, most prominently in stage14B and 14C (*Figure 6I* and *Figure 6—figure supplement 2A*). Mmp2::GFP formed a gradient that peaked at the posterior tip and gradually decreased toward the anterior. In contrast, there was marked reduction of Mmp2::GFP intensity in *hnt$^{RNAi}$* egg chambers (*Figure 6J–K* and *Figure 6—figure supplement 2B–C*). More than 80% of FC1- or FC2-expressing control egg chambers had moderate or high-level of Mmp2::GFP expression in their posterior follicle cells, while fewer than 30% of *hnt$^{RNAi}$* egg chambers (32% in the case of FC2 driving *hnt$^{RNAi2}$*) had moderate or high-level of Mmp2::GFP expression (*Figure 6L–M*). Due to technical challenges, we were unable to quantify Mmp2 protein level directly using western blotting. However, we speculated that Hnt might regulate *Mmp2* transcription. Therefore, we used real-time RT-PCR to quantify *Mmp2* mRNA level in control and *hnt$^{RNAi}$* egg chambers. Consistent with this hypothesis, *Mmp2* mRNA levels were significantly decreased in *hnt$^{RNAi}$* egg chambers in comparison to the control (*Figure 6N–O*). Altogether, these data demonstrate that Hnt regulates Mmp2 expression at the transcriptional level.

## Hindsight is required for *Oamb* expression in stage-14 follicle cells

We noticed that *hnt$^{RNAi2}$* egg chambers with FC2 Gal4 had slightly weaker reduction of *Mmp2* mRNA and protein expression (*Figure 6M and O*) and responded normally to ionomycin stimulation (*Figure 6H*), but were defective in OA-induced Ca$^{2+}$ rise, Mmp2 activation, and follicle rupture (*Figures 4H* and *5H*, and *Figure 6—figure supplement 1E–H*). This suggests that components upstream of Ca$^{2+}$ rise, for example Oamb, are defective in these egg chambers. Consistent with this hypothesis, *Oamb* mRNA was reduced two or more fold in *hnt$^{RNAi}$* egg chambers regardless the Gal4 drivers or *hnt$^{RNAi}$* lines (*Figure 7A–B*). Therefore, Hnt is also required for *Oamb* expression in stage-14 follicle cells.

Next, we aimed to rescue the rupture defect of *hnt$^{RNAi}$* egg chambers by overexpression of *Oamb*. *Oamb* overexpression was not able to restore the competency to OA-induced rupture in *hnt$^{RNAi}$* egg chambers with FC1 or *hnt$^{RNAi1}$* egg chambers with FC2, but it was able to do so in *hnt$^{RNAi2}$* egg chambers with FC2 (*Figure 7C–D*), consistent with the ionomycin experiment (*Figure 6D and H*). These data suggest that the major defect in *hnt$^{RNAi2}$* egg chambers with FC2 is the disruption of *Oamb* expression, while *hnt$^{RNAi1}$* egg chambers with FC2 or *hnt$^{RNAi}$* egg chambers with FC1 have more defects than Oamb alone, such as Mmp2 expression. The combination of later FC2 and weaker *hnt$^{RNAi2}$* may explain why only Oamb is majorly affected in this genetic manipulation. In addition, we noticed that egg chambers with *Oamb* overexpression alone initiated rupture before being able to perform the *ex vivo* culture (i.e. few intact egg chambers could be isolated). This is likely due to its high Oamb expression, which leads to high sensitivity to very low amount of endogenous OA released during egg chamber isolation. Nevertheless, all these data support the notion that Hnt transcriptionally upregulates Mmp2 expression in posterior follicle cells and then Oamb expression in all follicle cells to make stage-14 egg chambers to be competent to respond to OA-induced follicle rupture.

## Human RREB-1 can replace Hindsight's role in regulating follicle's competency to ovulation

To address whether the role of Hnt in stage-14 follicle cells can be replaced by its human homolog RREB-1, we first aimed to rescue the defects of *hnt$^{RNAi}$* egg chambers with *hnt* overexpression using *hnt$^{EP55}$* (see Materials and methods). To our surprise, overexpression of *hnt* in *hnt$^{RNAi}$* egg chambers did not rescue their defect in OA-induced follicle rupture (*Figure 8—figure supplement 1A and C*). In addition, these females laid similar numbers of eggs as *hnt$^{RNAi}$* females (*Figure 8—figure supplement 1B and D*). Surprisingly, Hnt protein was still depleted despite using FC1 or FC2 Gal4 driving

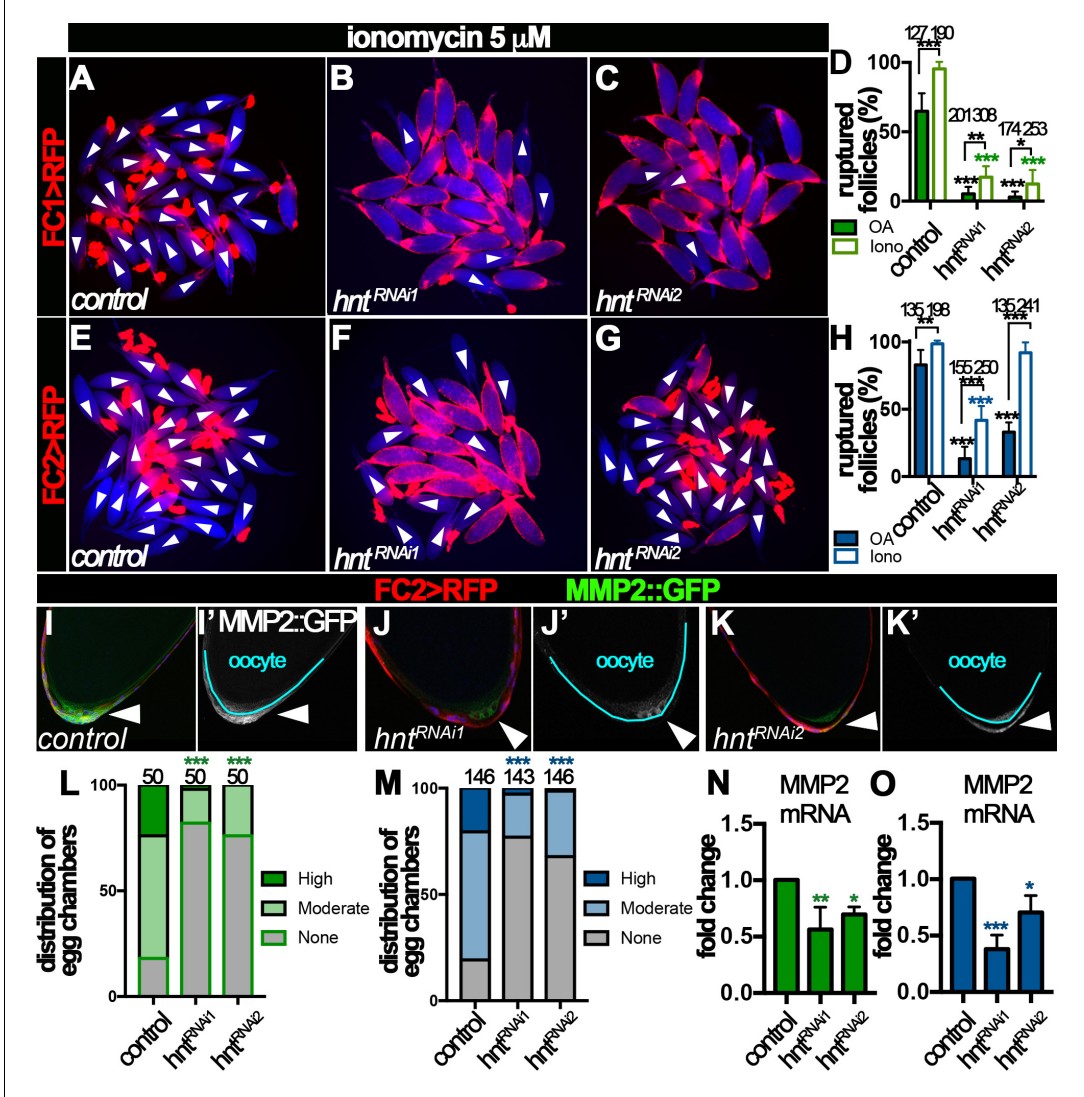

**Figure 6.** Hindsight regulates Mmp2 expression in stage-14 follicle cells. (A–H) Response of egg chambers isolated according to FC1 (A–D) or FC2 (E–H) to ionomycin-induced rupture in three hours. (A–C and E–G) Representative images show control (A and E) and *hnt*^RNAi^ (B–C and F–G) egg chambers after the culture. Bright field images of the egg chambers are shown in blue, and white arrowheads mark ruptured egg chambers. Quantification of rupture response to OA or to ionomycin is shown in (D and H). (I–K) Representative images show Mmp2::GFP expression (green in I–K and white in I'-K') in control (I–I') or *hnt*^RNAi^ (J–K') egg chambers with FC2 Gal4 (Red). Nuclei are labeled with DAPI and shown in blue (I–K). Arrowheads point to posterior follicle cells, and oocytes are outlined in cyan (I'–K'). (L–M) Quantification of Mmp2::GFP expression in control and *hnt*^RNAi^ egg chambers using FC1 (L) or FC2 (M) Gal4. (N–O) Quantification of *Mmp2* mRNA levels in *hnt*^RNAi^ egg chambers with FC1 (N) or FC2 (O) Gal4. *p<0.05, **p<0.01, ***p<0.001.

DOI: https://doi.org/10.7554/eLife.29887.012

The following figure supplements are available for figure 6:

**Figure supplement 1.** Measurement of intracellular Ca2$^{2+}$in follicle cells after ionomycin or octopamine stimulation.
DOI: https://doi.org/10.7554/eLife.29887.013
**Figure supplement 2.** *hnt* depletion disrupts Mmp2 expression in posterior follicle cells.
DOI: https://doi.org/10.7554/eLife.29887.014

*hnt*^EP55^ expression (**Figure 8—figure supplement 1E–L**), indicating that *hnt*^RNAi^ is sufficient to disrupt overexpressed *hnt* mRNA. This was further validated in a flip-out Gal4 system, in which Hnt protein was greatly reduced in cells with both *hnt*^RNAi^ and *hnt*^EP55^ (**Figure 8—figure supplement 1M–P**). Despite the failure to rescue ovulation in *hnt*-depleted females, it is interesting to note that overexpression of *hnt* alone with FC1 or FC2 Gal4 driver enhanced and suppressed OA-induced follicle

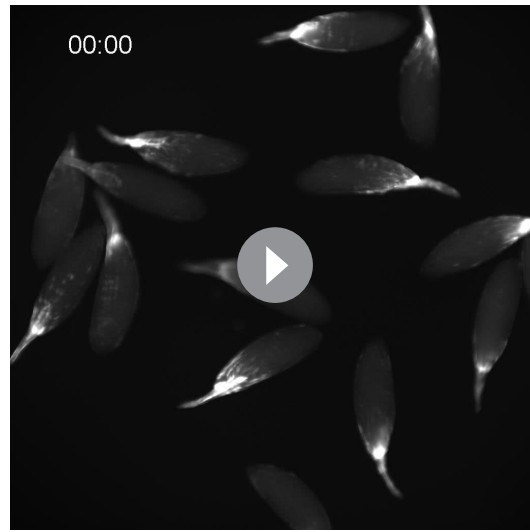

**Video 1.** Signal of GCaMP5G driven by FC2 in Control follicles with ionomycin stimulation (*FC2 > GCaMP5G*). DOI: https://doi.org/10.7554/eLife.29887.015

rupture, respectively (*Figure 8—figure supplement 1A and C*), suggesting that dynamic upregulation and downregulation of Hnt in stage-14 follicle cells may be required for normal function of these cells.

Next, a functional *RREB-1::GFP* fusion gene was overexpressed in *hnt^RNAi* females with FC1 to see whether RREB-1 could rescue the ovulation defect of *hnt^RNAi* females. *RREB-1* is successfully overexpressed in *hnt^RNAi* egg chambers, and overexpression of RREB-1 did not affect Hnt expression in control nor *hnt^RNAi* egg chambers (*Figure 8—figure supplement 2A–F*). *hnt^RNAi2*/ *RREB-1::GFP* females showed significant increase of egg-laying number in comparison to *hnt^RNAi2* females, indicating a rescue of ovulation defect (*Figure 8A*). This is supported by the result that *hnt^RNAi2*/*RREB-1::GFP* females spent 13 min, in comparison to 27 min in *hnt^RNAi2* females, to ovulate an egg, close to that in control females (*Supplementary file 1*). In contrast, females with *hnt^RNAi1*/*RREB-1::GFP* laid significantly fewer eggs than females with *hnt^RNAi1* alone (*Figure 8A*) and spent even longer time to ovulate an egg (*Supplementary file 1*). In addition, we noticed that these females frequently have eggs in the oviduct (*Supplementary file 1*), which may be caused by more frequent and uncoordinated follicle rupture leading to egg jamming in the oviduct. The persistence of egg in the oviduct may feedback to the ovary to inhibit further ovulation *in vivo*.

To more directly investigate the role of RREB-1 in ovulation, we isolated stage-14 egg chambers and performed OA-induced follicle rupture *ex vivo*. Excitingly, *RREB-1::GFP* overexpression was sufficient to rescue the rupture defect of *hnt^RNAi* egg chambers (*Figure 8B–F*), whereas overexpression of *UAS-GFP* was insufficient (*Figure 8—figure supplement 3*). In addition, overexpression of *RREB-1* alone led to increased OA-induced follicle

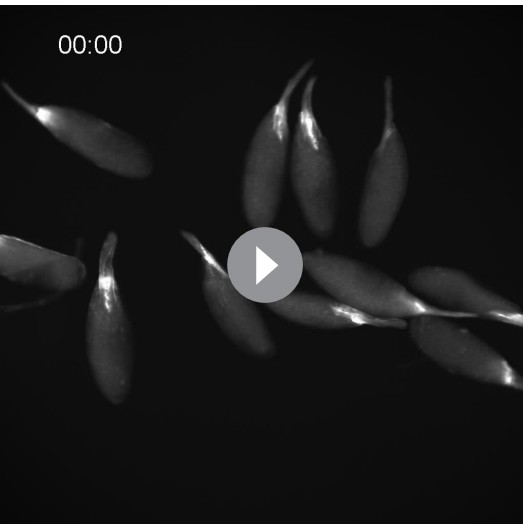

**Video 2.** Signal of GCaMP5G driven by FC2 in *hnt^RNAi1* follicles with ionomycin stimulation (*FC2 > GCaMP5G/ hnt^RNAi1*). DOI: https://doi.org/10.7554/eLife.29887.016

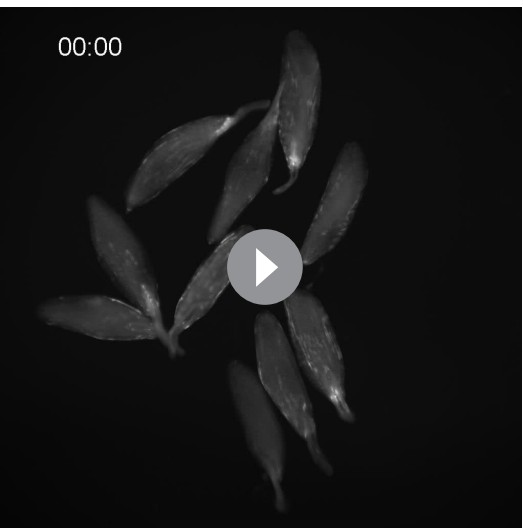

**Video 3.** Signal of GCaMP5G driven by FC2 in *hnt^RNAi2* follicles with ionomycin stimulation (*FC2 > GCaMP5G/ hnt^RNAi2*). DOI: https://doi.org/10.7554/eLife.29887.017

rupture, similar to overexpression of *hnt* with FC1 (*Figure 8B* and *Figure 8—figure supplement 1A*). Consistent with the rescue of follicle rupture, both *Mmp2* and *Oamb* mRNA expression was rescued to normal or even higher level by overexpression of *RREB-1* (*Figure 8G–H*). Therefore, RREB-1 can replace Hnt's role in upregulating Mmp2 and Oamb expression in follicle cells. Altogether, our data demonstrate that zinc-finger transcription factor Hnt/RREB-1 may play conserved roles in promoting follicle maturation and ovulation competency.

## Discussion

### Hindsight regulates ovulation competency in stage-14 egg chambers

Work in this study demonstrated for the first time that Hnt has a dynamic expression pattern in stage-14 follicle cells and is a key factor for the final maturation of stage-14 egg chambers

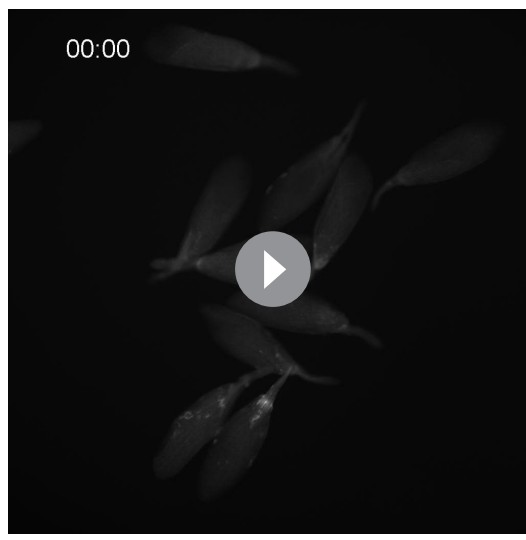

**Video 4.** Signal of GCaMP5G driven by FC2 in Control follicles with octopamine stimulation (*FC2 > GCaMP5G*). DOI: https://doi.org/10.7554/eLife.29887.018

(*Figure 9*). Oocyte maturation has been well studied in *Drosophila* and other species (*Eichhorn et al., 2016*; *Kronja et al., 2014*; *Von Stetina and Orr-Weaver, 2011*); however, the maturation of follicle cells surrounding the oocyte in the stage-14 egg chamber is poorly defined at the molecular level (*Duhart et al., 2017*; *Klusza and Deng, 2011*; *Spradling, 1993*). According to Hnt expression in stage-14 egg chambers, we define the stage-14 egg chambers into three sub stages. Hnt is first upregulated in posterior follicle cells of stage-14A egg chambers, which is likely corresponding to Hnt's role in upregulating Mmp2 expression in these follicle cells (*Figure 9*). Then Hnt is upregulated in all main-body follicle cells of stage-14B egg chambers, which is likely corresponding to Hnt's role in upregulating *Oamb* expression (*Figure 9*). The sequential upregulation of Mmp2 and Oamb is fully consistent with the fact that FC1-expressing egg chambers, in comparison to FC2-

expressing egg chambers, are less efficient for OA-induced follicle rupture, but fully competent to respond to ionomycin-induced follicle rupture (*Figures 4D, H*, *6D and H*). The orchestrated upregulation of Mmp2 and Oamb, and possibly other components in the OA/Oamb-Ca$^{2+}$-Mmp2 pathway, by Hnt makes the final stage-14C egg chambers fully competent for ovulation. Components in the ecdysteroid signaling pathway, including the enzyme Shd for steroid production and Ecdysone receptor (EcR), also changes its expression pattern from stage-13 to stage-14 (*Knapp and Sun, 2017*). It is unknown whether Hnt is also responsible for such changes; however, it is unlikely that ecdysteroid signaling upregulates Hnt in stage-14 follicle cells, because ecdysteroid signaling does not affect Mmp2 and Oamb expression (*Knapp and Sun, 2017*). Hnt is upregulated in follicle cells from stage 7 to stage 10A, which depends on Notch signaling (*Sun and Deng, 2007*); however, Notch signaling is not active in stage-14 follicle cells and is unlikely to upregulate Hnt at this stage.

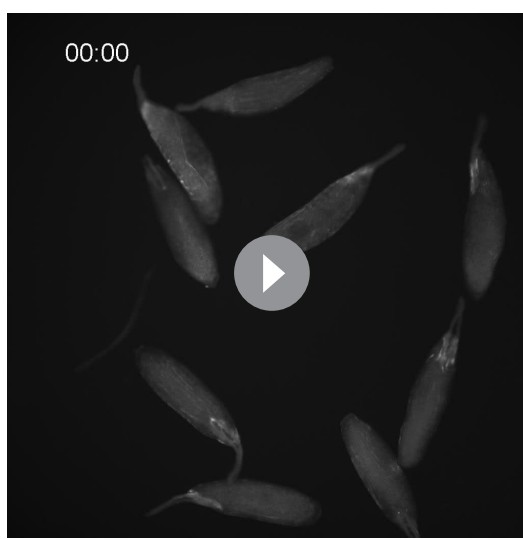

**Video 5.** Signal of GCaMP5G driven by FC2 in *hnt*[RNAi1] follicles with octopamine stimulation (*FC2 > GCaMP5G/hnt*[RNAi1]). DOI: https://doi.org/10.7554/eLife.29887.019

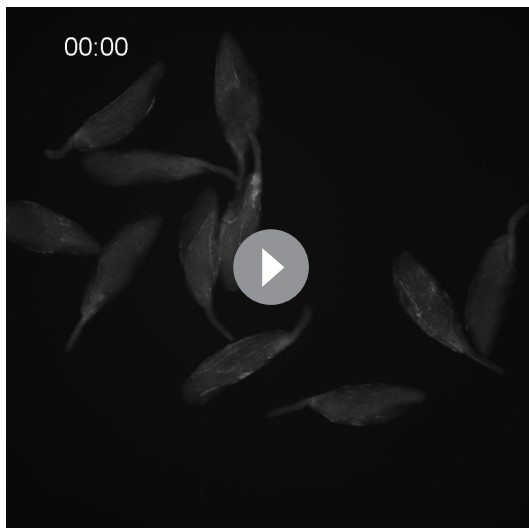

**Video 6.** Signal of GCaMP5G driven by FC2 in *hnt*^RNAi2^ follicles with octopamine stimulation (*FC2 > GCaMP5G/hnt*^RNAi2^).
DOI: https://doi.org/10.7554/eLife.29887.020

Thus, the developmental signal for Hnt upregulation in stage14 and the transition from stage 13 to stage 14 is still unknown.

## Hindsight and its role in extracellular matrix homeostasis and epithelial integrity

*Mmp2*, along with *Mmp1*, are the only genes in the fly genome encoding matrix metalloproteinase and are crucial for extracellular matrix homeostasis during normal development, wound repair, and cancer metastasis (*Page-McCaw, 2008*; *Stevens and Page-McCaw, 2012*). Unlike Mmp1, whose expression is tightly regulated by Jun-related kinase (JNK) signaling (*Uhlirova and Bohmann, 2006*), regulation of Mmp2 expression is largely unknown. Our work clearly defines the role of Hnt in regulating Mmp2 expression and basement membrane remodeling during ovulation. Hnt directly binds to two adjacent Hnt-binding sequences in the regulatory region of *hnt* and *nervy* genes and attenuates their expression (*Ming et al., 2013*). Such Hnt-binding motifs are not found in the gene region of *Mmp2 and Oamb*. Thus, Hnt may indirectly regulate Mmp2 expression in posterior follicle cells. In addition, other transcriptional regulators must exist to coordinate with Hnt to restrict Mmp2 expression to posterior follicle cells.

Hnt's role in regulating Mmp2 expression and extracellular matrix homeostasis may not be restricted to posterior follicle cells. It has been shown that Hnt has a general role in regulating epithelial integrity in multiple organ systems and developmental stages. During retinal morphogenesis, *hnt* mutant photoreceptor cells frequently delaminate from retinal epithelium and are unable to maintain their integrity (*Pickup et al., 2002*). In the tracheal system, *hnt* mutant tracheal epithelium disintegrate to form sacs and vesicles from collapsed dorsal trunk and branches (*Wilk et al., 2000*). During oogenesis, Hnt is essential for proper cell adhesion and collective cell migration in stage-9 egg chambers. Ectopic expression of Hnt in the cluster of border cells leads to dissociation of the border-cell cluster (*Melani et al., 2008*). In addition, genetic modifier screens identify basement-membrane components Vkg and Laminin as Hnt's genetic interactors (*Wilk et al., 2004*). All of these studies suggest that Hnt plays general roles in regulating epithelial integrity and extracellular matrix homeostasis in multiple organ systems. It will be interesting to see whether the regulation of Mmp2 by Hnt also occurs in other Hnt-expressing or Mmp2-expressing tissues/organs.

## Hindsight and human RREB-1 are functionally conserved in ovulation

*Drosophila* Hnt and mammalian RREB-1 are functionally conserved in many aspects. Both *Drosophila* Hnt and mammalian RREB-1 are required for proper cell migration (*Melani et al., 2008*). Human RREB-1 binds to similar DNA sequences in *Drosophila* salivary gland polytene chromosomes as Hnt and rescues the germ band retraction phenotype in *hnt* mutant embryos (*Ming et al., 2013*). In addition, our current work shows that human RREB-1 is able to rescue *Oamb* and *Mmp2* expression in stage-14 follicle cells and OA-induced follicle rupture/ovulation phenotype in *hnt*^RNAi^ females (*Figure 8*). The role of RREB-1 in mammalian ovulation has not been studied so far, however, RREB-1 is detected in granulosa cells in mouse ovaries by microarray analysis (*Fan et al., 2009*). In addition, mammalian RREB-1 functions downstream of the Ras-MAPK signaling pathway in multiple occasions (*Kent et al., 2010*; *2013*; *Mukhopadhyay et al., 2007*; *Thiagalingam et al., 1996*; *Zhang et al., 2003*), and the Ras-MAPK signaling pathway is involved in mammalian ovulation (*Fan et al., 2009*). It is possible that RREB-1 may function in granulosa cells to regulate Mmp expression and ovulation downstream of Ras-MAPK pathway in mammals.

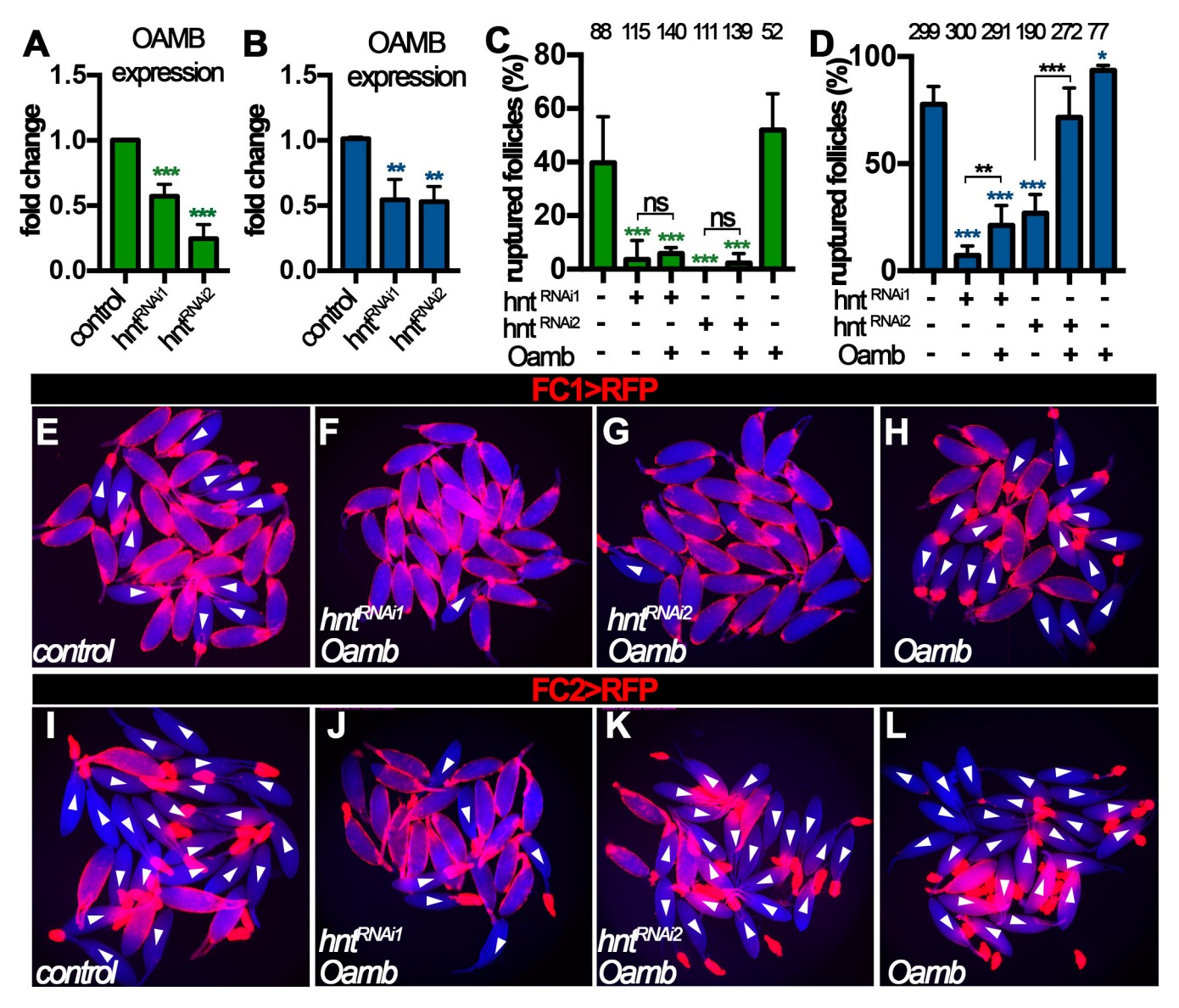

**Figure 7.** Hnt regulates *Oamb* expression in stage-14 follicle cells. (A–B) Quantification of *Oamb* mRNA levels in *hnt*[RNAi] egg chambers with FC1 (A) or FC2 (B) Gal4. (C–D) Quantification of egg chambers in response to OA-induced follicle rupture. *hnt*[RNAi] and/or *Oamb* overexpression is driven by FC1 (C) or FC2 (D) Gal4. (E–L) Representative images of the quantification in (C–D). FC1 >RFP (E–H) or FC2 >RFP (I–L) is shown in red, bright-field images of the egg chambers are shown in blue, and ruptured egg chambers are marked by white arrowheads. *p<0.05, **p<0.01, ***p<0.001.
DOI: https://doi.org/10.7554/eLife.29887.021

## Materials and methods

### *Drosophila* genetics

Flies were reared on standard cornmeal and molasses food at 25°C, and all RNAi-mediated depletion experiments were performed at 29°C with *UAS-dcr2*. Two stage-14 follicle-cell specific Gal4 drivers from the Janelia Gal4 collection (*Pfeiffer et al., 2008*) were used in this study: *R44E10-Gal4* (FC1) and *R47A04-Gal4* (FC2). The following RNAi lines were used: *UAS-hnt*[RNAi1] (V3788) and *UAS-hnt*[RNAi2] (V101325) from the Vienna *Drosophila* Resource Center; and *UAS-Mmp2*[RNAi] (*Uhlirova and Bohmann, 2006*). *UAS-Oamb.K3* (*Lee et al., 2009*), *UAS-Timp* (*Page-McCaw et al., 2003*), *hnt*[EP55] (a P-element insertion line containing UAS sequence in the promoter region of *hnt*; Bloomington

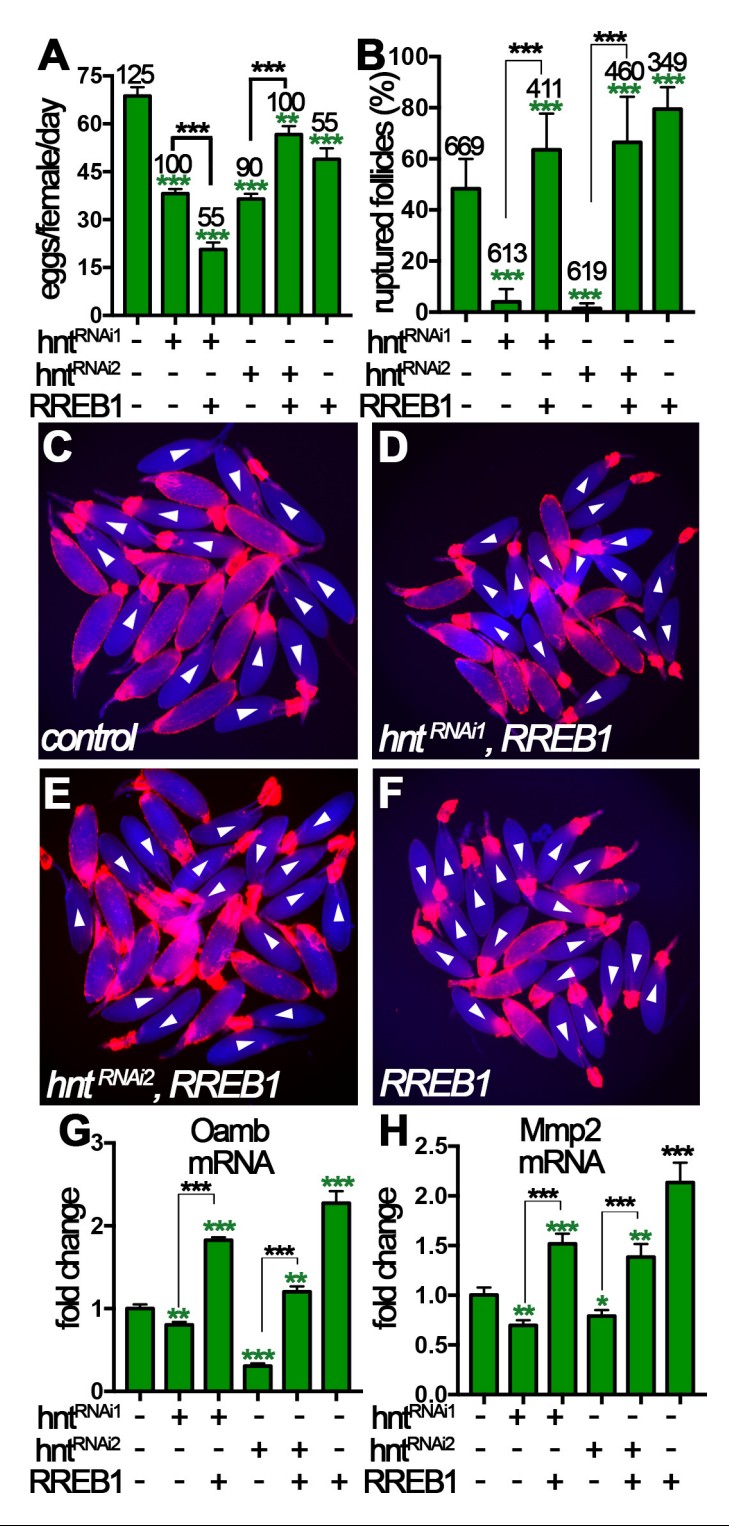

**Figure 8.** Human RREB-1 can replace Hindsight's role in regulating follicle's competency to ovulation. (**A**) The quantification of egg-laying capacity of females with FC1 driving *hnt^RNAi* and/or *RREB-1* overexpression. (**B–F**) The quantification of OA-induced follicle rupture (**B**) in egg chambers with *hnt^RNAi* and/or *RREB-1* overexpression using FC1 Gal4. Representative images are shown in (**C–F**). FC1 >RFP is shown in red, bright-field images of egg chambers are shown in blue, and white arrowheads mark ruptured follicles. (**G–H**) Quantification of *Oamb* (**G**) and *Mmp2* (**H**) mRNA level in egg chambers with FC1 Gal4 driving *hnt^RNAi* and/or *RREB-1* overexpression. *p<0.05, **p<0.01, ***p<0.001.

*Figure 8 continued on next page*

*Figure 8 continued*
DOI: https://doi.org/10.7554/eLife.29887.022
The following figure supplements are available for figure 8:
**Figure supplement 1.** Rescue of ovulation defect in *hnt^{RNAi}* females with *hnt* overexpression.
DOI: https://doi.org/10.7554/eLife.29887.023
**Figure supplement 2.** Human RREB-1::GFP does not interfere with Hnt expression in control or *hnt^{RNAi}* egg chambers.
DOI: https://doi.org/10.7554/eLife.29887.024
**Figure supplement 3.** Overexpression of GFP is not sufficient to rescue rupture defect of *hnt^{RNAi}* follicles with FC1 Gal4.
DOI: https://doi.org/10.7554/eLife.29887.025

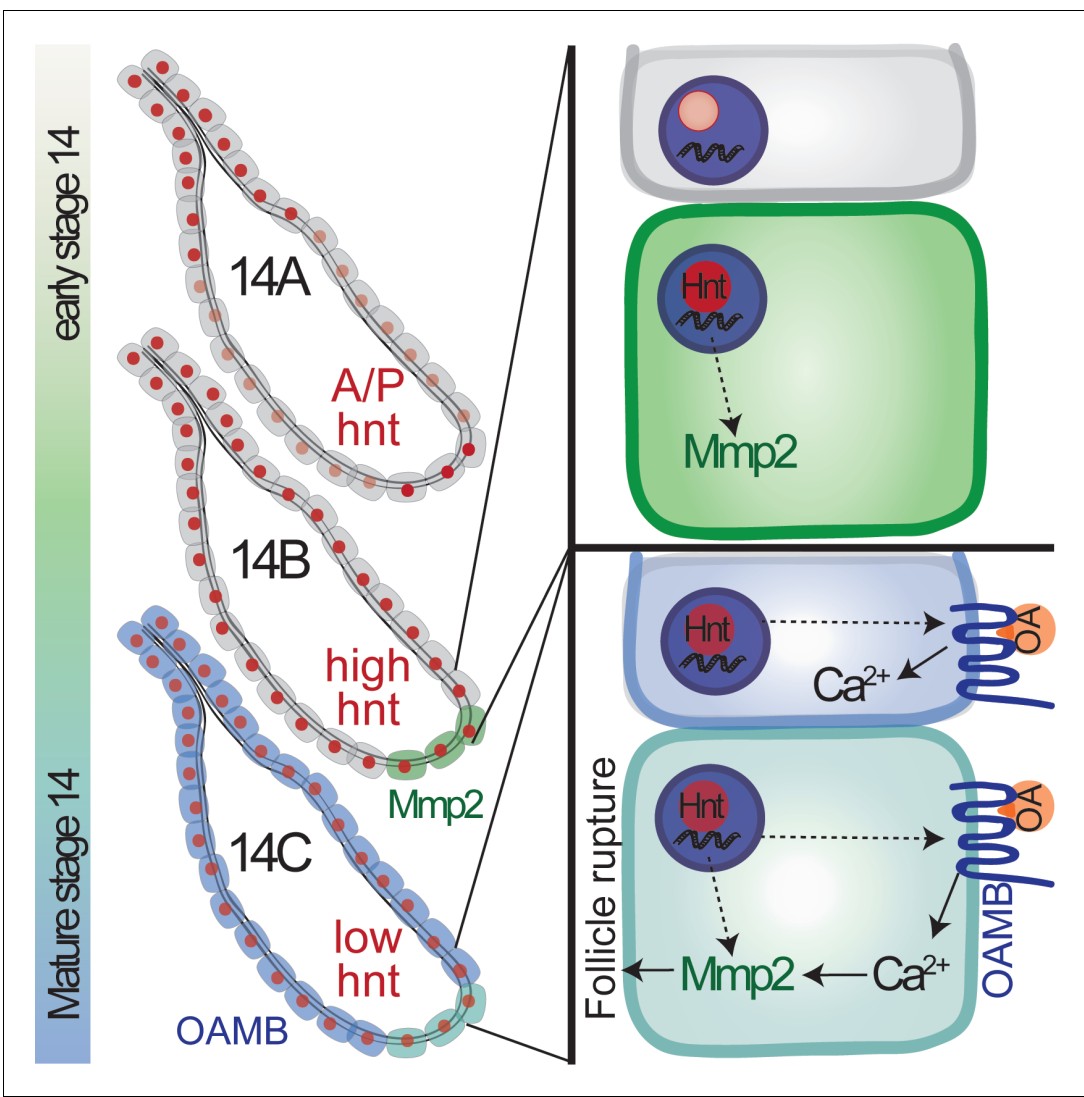

**Figure 9.** A schematic cartoon summarizes the role of Hindsight in stage-14 follicle cells. Hnt expression is shown in red with different intensity indicating different expression level. Mmp2 expression is shown in green and Oamb expression is shown in blue. OA stands for octopamine.
DOI: https://doi.org/10.7554/eLife.29887.026

Drosophila Stock Center, BDSC# 5358), *UAS-RREB1::GFP* (*Ming et al., 2013*) were used to overex-press *Oamb*, *Timp*, *Hnt*, and *RREB1*, respectively. *Oamb.K3* is the *Oamb* isoform expressed in wild-type stage-14 follicle cells (*Deady and Sun, 2015*). *hnt$^{XE81}$* and *hnt$^{EH704a}$* are loss-of-function *hnt* alleles, while *hnt$^{peb}$* (BDSC# 80) is a temperature-sensitive *hnt* allele (*Wilk et al., 2004*). Animals bearing *hnt$^{peb}$* were raised at room temperature, and newly emerged adult flies were shifted to the 29°C restrictive temperature. For generating flip-out actin-Gal4 clones (*Pignoni and Zipursky, 1997*), hsFLP;;act <CD2<Gal4,UAS-RFP/TM3, Sb (derived from BDSC# 30558) was used to cross to *hnt$^{EP55}$* or *hnt$^{EP55}$; UAS-hnt$^{RNAi}$* and adult flies were heat shocked in a 37°C water bath for 45 min. *UASp-GFP::act79B; UAS-mCD8::GFP* was crossed to Gal4 lines and used to visualize Gal4 expression pattern. *UAS-RFP* was recombined to Gal4 drivers and used for isolating stage-14 egg chambers for *ex vivo* culture. *UAS-GCaMP5G* (*Akerboom et al., 2012*) was used to visualize calcium responses in follicle cells (BDSC# 42037). Protein trap lines *vkg::GFP$^{CC00791}$* (*Buszczak et al., 2007*) and *Mmp2::GFP* (*Deady et al., 2015*) were used for Vkg and Mmp2 expression, respectively. Control flies for all experiments were prepared from crossing Gal4 drivers to Oregon-R.

## Ovulation assays

Egg laying and egg-laying time analyses were performed as previously described (*Deady and Sun, 2015*; *Knapp and Sun, 2017*). Five virgin females (five-to-six days old with one day of wet yeast feeding) were placed with ten Oregon-R males in one bottle to lay eggs on grape juice-agar plates with a drop of wet yeast paste for two days in 29°C. After each day (22 hr in 29°C) of egg laying, grape juice-agar plates were removed and replaced with a new one. Typically, five bottles for each genotype are performed in each experiment. After egg laying, ovaries were dissected and mature follicles in female ovaries were counted. Virgin females were dissected before mating for a 'pre-egg laying' mature follicle count to ensure normal oogenesis occurred. The average number of eggs laid per female per day was used to calculate the average time to lay one egg, as described previously. The egg-laying time was further proportioned into the amount of time an egg spent in the ovary (ovulation time), in the oviduct (oviduct time), and in the uterus (uterus time) according to the distribution of females with eggs in their reproductive tract six hours after mating. For this assay, ten virgin females and fifteen males are mated in a vial with dry yeast at 29°C. Typically, two to three vials for each genotype were performed in each experiment. After a six-hour mating, the flies were frozen at −80°C for approximately four minutes, and then dissected to examine the location of an egg within the reproductive tract.

## *Ex vivo* follicle rupture, gelatinase assay, and quantitative RT-PCR

The *ex vivo* follicle rupture assay was performed as described previously (*Deady and Sun, 2015*). In brief, 5–6 day-old virgin females fed with wet yeast for 2–3 days were used to isolate stage-14 egg chambers in Grace's insect medium (Caisson Lab, Smithfield, UT). Within one hour, isolated mature follicles from ~10 females were separated into groups of ~30 egg chambers, then cultured in culture media (Grace's medium, 10% fetal bovine serum, and 1X penicillin/streptomycin) supplemented with 20 μM OA (Sigma, St. Louis, MO), or 5 μM ionomycin (Cayman Chemical Co., Ann Arbor, MI). All cultures were performed at 29°C, the same condition as flies were maintained, to enhance Gal4/UAS efficiency. One data point represents the percent of ruptured follicles per experimental group (~30 egg chambers). Data were represented as mean percentage ± standard deviation (SD).

In situ zymography for detecting gelatinase activity was performed as previously reported with minor modifications (*Deady and Sun, 2015*). 20–25 μg/mL of DQ-gelatin conjugated with fluorescein (Invitrogen, Eugene, OR) was added into the culture media with 20 μM OA for three hours. Mature follicles with posterior fluorescein signal were directly counted, and data represented as percent of follicles with posterior fluorescein signal. Follicles with lateral fluorescein signal, which is likely induced by damage during dissection, are not counted as Mmp2 activity, because Mmp2 is only expressed in posterior follicle cells (*Deady et al., 2015*).

For quantitative RT-PCR, total RNA was extracted from 60 stage-14 egg chambers isolated from 10 flies using Direct-zol RNA MicroPrep Kit (Zymo Research, Orange, CA). cDNA synthesis, real-time PCR amplification and primers of *Oamb.K3* and *Mmp2* were described previously (*Knapp and Sun, 2017*). The data are presented as mean ± SEM from three biological replicates, except for *RREB-1* rescue experiment, in which one single biological experiment was presented.

## Immunostaining and microscopy

Immunostaining was performed following a standard procedure, including ovary dissection, fixation in 4% EM-grade paraformaldehyde for 15 min, blocking in PBTG (PBS + 0.2% Triton + 0.5% BSA + 2% normal goat serum), and primary and secondary antibody staining diluted in PBTG. For *vkg::GFP* analysis, stage-14 egg chambers were first isolated from ovaries in cold Grace's medium before fixation. Mouse anti-Hnt (1:75; Developmental Study Hybridoma Bank), mouse anti-GFP (1:2000; Invitrogen), rabbit anti-GFP (1:4000; Invitrogen), and rabbit anti-RFP (1:1000; MBL international) were used as primary antibodies, and Alexa 488, 546, and 633 goat anti-mouse and goat anti-rabbit (1:1000, Invitrogen) were used as secondary antibodies. Images were acquired using a Leica TCS SP8 confocal microscope or Leica MZ10F fluorescent stereoscope with a sCOMS camera (PCO.Edge), and assembled using Photoshop software (Adobe Inc., Mountain View, CA) and ImageJ.

To visualize calcium response to ionomycin and octopamine, egg chambers expressing GCaMP5G and *hnt^{RNAi}* were isolated into an imaging chamber. Images were acquired on a Zeiss Axio Zoom microscope at 0.2 FPS, and 10 μL of ionomycin or octopamine were added to the solution after frame five to a final concentration of 5 μM or 20 μM, respectively. A ROI in the center of the main-body follicle cells was selected and the integrated intensity was measured. F0 was defined as the average baseline intensity (first five frames), and ΔF/F0 is reported.

## Statistical analysis

Statistical tests were performed using Prism 7 (GraphPad, San Diego, CA). For comparison of more than two means, one-way ANOVA with *post hoc* Fisher's Least Significant Difference test was used. For comparison of distribution, Chi square test was used except in *Figure 3D and H*, where Fisher's exact test was used. In addition, Z-score test was used for egg-laying time analysis in *Figure 3K–L* and *Supplementary file 1*.

# Acknowledgements

We are thankful to Drs. Howard Lipshitz, Bruce Reed, Allan Spradling, Dirk Bohmann, Kyung-An Han, Andrea Page-McCaw, and Wu-Min Deng for sharing fly lines, the Bloomington Drosophila Stock Center and the Vienna Drosophila Stock Center for fly stocks, and the DHSB for antibodies. We are very grateful to Virge Kask at UConn Scientific Illustration Office for providing the illustration of *Drosophila* ovulation process, to the Conover, LoTurco, and Kanadia labs for sharing reagents and microscopes. We thank Drs. John Peluso and Joseph LoTurco as well as the anonymous reviewers for discussion and comments regarding the manuscript; and are grateful for Elizabeth Knapp and Wei Shen for technical support. Leica SP8 confocal microscope is supported by a NIH award (S10OD016435) to Akiko Nishiyama. JS is supported by the University of Connecticut Start-up fund, NIH/National Institute of Child Health and Human Development Grant R01-HD086175, and Bill and Melinda Gates Foundation.

# Additional information

## Funding

| Funder | Grant reference number | Author |
| --- | --- | --- |
| National Institutes of Health | R01-HD086175 | Jianjun Sun |
| Bill and Melinda Gates Foundation | OPP1160858 | Jianjun Sun |
| University of Connecticut | Startup | Jianjun Sun |

The funders had no role in study design, data collection and interpretation, or the decision to submit the work for publication.

## Author contributions
Lylah D Deady, Formal analysis, Investigation, Writing—original draft, Writing—review and editing; Wei Li, Formal analysis, Investigation, Writing—review and editing; Jianjun Sun, Conceptualization, Formal analysis, Supervision, Funding acquisition, Project administration, Writing—review and editing

## Author ORCIDs
Lylah D Deady http://orcid.org/0000-0002-9316-1326
Wei Li http://orcid.org/0000-0001-8699-5325
Jianjun Sun http://orcid.org/0000-0002-6015-738X

## Decision letter and Author response
Decision letter https://doi.org/10.7554/eLife.29887.030
Author response https://doi.org/10.7554/eLife.29887.031

## Additional files

### Supplementary files
• Supplementary file 1. The egg laying, egg distribution within the reproductive tract, and egg-laying time of females with various genotypes.
DOI: https://doi.org/10.7554/eLife.29887.027

• Transparent reporting form
DOI: https://doi.org/10.7554/eLife.29887.028

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
