## [Decision Letter]

Thank you for submitting your article "Zinc-finger transcription factor Hindsight regulates ovulation competency of *Drosophila* follicles" for consideration by *eLife*. Your article has been reviewed by two peer reviewers, one of whom is a member of our Board of Reviewing Editors, and the evaluation has been overseen by Marianne Bronner as the Senior Editor.

The reviewers have discussed the reviews with one another and the Reviewing Editor has drafted this decision to help you prepare a revised submission.

Summary:

The manuscript "Zinc-finger transcription factor Hindsight regulates ovulation competency of *Drosophila* follicles"*eLife* by Deady, Li, and Sun is a thoroughly conducted study of what is possibly a conserved mechanism regarding the release of mature oocytes from the ovary. The present study builds on previous work and explores the regulation of oocyte competency in the *Drosophila* system. The submitted manuscript establishes that the expression of hindsight (hnt), the *Drosophila* ortholog of mammalian RREB-1, is required for this process. The manuscript is well written and the results are, for the most part, clearly presented.

Using follicle cell-specific GAL4 drivers as well as the morphology of nurse cell degeneration, the authors report that developing stage 14 egg chambers show strong hnt expression in follicle cells in a highly dynamic pattern. Stage 14 egg chambers are subdivided into 14A, 14B, and 14C stages and, stage 14B is shown to have the highest level of hnt expression. RNAi mediated knockdown of Hnt in stage 14 egg chambers is shown to result in ovulation defects and not oogenesis defects. These defects are demonstrated as being associated with reduced Mmp2 activity using a variety of assays. The hnt-knockdown phenotypes are corroborated by expression of the Mmp2 inhibitor Timp as well as Mmp2 RNAi mediated knockdown. Using an ex vivo assay, hnt-knockdown stage 14 egg chambers were found to be unresponsive to treatment with the Ca^2+^ ionophore ionomycin. Later stage, weaker hnt-knockdown egg chambers were responsive to ionomycin but were not responsive to the OA ligand for Oamb, which was previously known to be required for ovulation. The authors go on to show that reduced hnt expression results in reduced *Mmp2* and Oamb expression, and correctly state that this is not necessarily a direct relationship. Based on RT-PCR the authors show that hnt likely functions to up-regulate the transcription of Mmp2 and Oamb. It is suggested that RNAi mediated hnt knockdown can be rescued by co-expression of human RREB1-GFP.

Overall, this work is sound and is certainly of great interest to *Drosophila* oogenesis specialists. The extent to which this work would be of broader interest relates to the final rescue experiment, and it is here we feel further experiments would be required to support and validate the idea of a conserved function of hnt in ovulation.

Essential revisions:

Although most of the experiments are conducted in a rigorous manner, supporting their conclusions, weaknesses revolve around the fact that the validation of the RNAi-based method is not sufficient to fully support their conclusions. First, the data in Figure 2—figure supplement 1 show that knockdown is quite incomplete. Thus, additional validation that the phenotypes are indeed caused by hnt knockdown is required.

The *UAS-RREB-1* rescue experiment should be controlled for possible GAL4 / UAS "competition". This key experiment shows that there is a rescue of the hnt RNAi mediated knockdown phenotype by co-expression of *UAS-RREB-1::GFP*. I believe that it is possible that the rescue effect could be due to reduced efficiency of the RNAi knockdown, possibly through a non-specific effect such as less effective induction of UAS-hnt-RNAi expression due to the presence of the additional UAS responder element of *UAS-RREB-1::GFP*. (A possible titration effect of having additional UAS sequences available for the GAL4.) To control for this possibility of "GAL4 competition", a control experiment should be performed with a UAS-GFP in place of *UAS-RREB-1::GFP*. The level of expression of the UAS-GFP should be comparable to the *UAS-RREB-1::GFP*.

Does hnt-overexpression resemble RREB-1 expression? It would be informative to show if there is a phenotype associated with hnt over-expression (using UAS-hnt, UAS-GFP-Hnt, EP55, or P^GSV1^GS1018) in stage 14 follicle cells. Or if not over-expressed per se, is there a consequence to a prematurely high level of hnt expression, or maintenance of a high level of expression, in stage 14C – and could this address whether the down-regulation of hnt expression in 14C stage is functionally significant? I.e. Is hnt over-expression sufficient for the premature rupture phenotype and/or upregulation of *Mmp2* and Oamb? Would such a phenotype resemble UAS-RREB1 expression, for example? These are fairly achievable experiments and their omission from this study seems puzzling to me.

[Editors' note: further revisions were requested prior to acceptance, as described below.]

Thank you for resubmitting your work entitled "The zinc-finger transcription factor Hindsight regulates ovulation competency of *Drosophila* follicles" for further consideration at *eLife*. Your revised article has been favorably evaluated by Marianne Bronner (Senior editor), a Reviewing editor, and one reviewer.

The manuscript has been improved but there are some remaining issues, described in the review attached below, that need to be addressed before acceptance. These remaining issues are minor and can be addressed easily, so we anticipate to be able to accept your manuscript without further review upon revision. If you choose not to do the requested experiment, please provide an explanation in your response letter.

Reviewer #2:

I have gone through the revised manuscript, " The zinc-finger transcription factor Hindsight regulates ovulation competency of *Drosophila* follicles"*eLife* by Deady, Li, and Sun.

I believe the authors have made clarifications and explanations to my satisfaction. It is an interesting story – and the investigations of ovulation and follicle cell rupture as a conserved mechanism is of general interest to the field of developmental biology.

I believe that much of the difficulty relates to the subtly with which hnt depletion or expression can affect the phenotype during the relatively short time window of stage 14 egg chambers.

I was initially concerned that the RREB-1 rescue of UAS-hnt-RNAi might not be specific. The authors did address my concerns regarding UAS-titration effects.

The authors spend a fair amount of time addressing the issue of using GAL4 mediated co-expression of UAS-hnt-RNAi and UAS-hnt, presumably to demonstrate that the UAS-hnt-RNAi phenotype is specific to hnt depletion. I'm not sure if these arguments, since the experiments were not effective, add anything to the manuscript other than some confusion.

What was much more convincing in my mind was the demonstration of ovulation defects using the temperature sensitive allele *hnt^peb^* – which the authors call hnt^1^, as shown in the supplement to Figure 4. In the follicle cell rupture, both *hnt^peb^/hnt^XE81^* and *hnt^peb^/hnt^EH704a^* were reduced to below 30%, where FC2 controls in Figure 4 show ~80%.

Two questions come to mind:first,is *hnt^peb^* /+ control (~ 50%) showing a dosage effect compared to + /+ control (~80%)?;

and second,can the loss-of-function phenotype of *hnt^peb^/hnt^XE81^* and *hnt^peb^* /*hnt^EH704a^* be rescued by FC1 or FC2 GAL4 expression of *UAS-RREB-1*?

This latter experiment would completely and definitively address any concerns regarding effectiveness of RNAi depletion phenotypes and their rescue by UAS-RREB1.

Since the rescue of the knockdown is pivotal to a conserved function, I would really like to see inclusion of this experiment. It should only take a couple of fly generations to get the GAL4 + UAS lines into the *hnt^peb^/hnt^XE81^* and *hnt^peb^/hnt^EH704a^* backgrounds, and the assay is well established in the lab. I think it will strengthen the rescue results.

Also, is there any evidence that RREB-1 is expressed in the mammalian ovary? Is it possible to find out somehow? (Such as some high throughput data somewhere?) If so, then it would certainly strengthen the conserved function in ovulation. But if it is not? Then…?

---

## [Author Response]

Essential revisions:Although most of the experiments are conducted in a rigorous manner, supporting their conclusions, weaknesses revolve around the fact that the validation of the RNAi-based method is not sufficient to fully support their conclusions. First, the data in Figure 2—figure supplement 1 show that knockdown is quite incomplete. Thus, additional validation that the phenotypes are indeed caused by hnt knockdown is required.

In this study, we have used two independent *hnt^RNAi^* lines, which are targeting non-overlapping regions of *hnt* mRNA (*hnt^RNAi1^* targeting nucleotide 4864-5206 and *hnt^RNAi2^* targeting nucleotide 1787-2077), to demonstrate that both *hnt^RNAi^* lines cause similar ovulation phenotypes. Although both *RNAi* lines are predicted to have potential off-targets (*hnt^RNAi1^*: CG10597 and CG5119; *hnt^RNAi2^*: CG10228, CG12673, and CG7991), the only overlapping target between these two *hnt^RNAi^* lines is the *hnt* gene. These data strongly suggest that *hnt* is responsible for the ovulation defect. We have also tried to use *UAS-hnt^EP55^* to rescue the defect of *hnt^RNAi^* females. Despite using the Gal4/UAS system to overexpress *hnt* mRNA, unfortunately Hnt protein was still depleted by RNAi and not restored to normal level, and thus no rescue was observed (we added this result to new Figure 8—figure supplement 1). In contrast, mammalian homolog RREB-1, which will not be targeted by *hnt^RNAi^*, is able to fully rescue the follicle rupture defect caused by *hnt* knockdown (Figure 8). This data also strongly support our conclusion that rupture defect in RNAi-based method is indeed caused by *hnt* knockdown, not peculiar RNAi off-target effect. Furthermore, we showed that mature follicles from *hnt* transheterozygous mutant females with a temperature-sensitive *hnt* allele also showed defect in OA-induced follicle rupture (Figure 4—figure supplement 1). All these genetic evidence support our conclusion that *hnt* is the gene required for follicle rupture/ovulation.

Reviewers also pointed out that *hnt^RNAi^* knockdown is quite incomplete. It is true that *hnt* knockdown is incomplete, which is largely due to the late expression of Gal4 driver (both are in stage-14 egg chambers) and short developmental time (stage 14 only lasts for a little more than two hours and it only takes six hours to develop from stage 10 to stage 14). However, *hnt* is strongly depleted whenever Gal4 driver has high expression. For example, FC1 Gal4 is efficient to deplete *hnt* in stage 14B and stage 14C (Figure 3 and Figure 3—figure supplement 1), while FC2 Gal4 is efficient to deplete *hnt* in stage 14C (Figure 3). The incomplete depletion of *hnt* is also quite consistent with incomplete block of ovulation in *hnt^RNAi^* females. Thus, we strongly believe the ovulation defect is indeed caused by *hnt* depletion.

The UAS-RREB-1 rescue experiment should be controlled for possible GAL4 / UAS "competition". This key experiment shows that there is a rescue of the hnt RNAi mediated knockdown phenotype by co-expression of UAS-RREB-1::GFP. I believe that it is possible that the rescue effect could be due to reduced efficiency of the RNAi knockdown, possibly through a non-specific effect such as less effective induction of UAS-hnt-RNAi expression due to the presence of the additional UAS responder element of UAS-RREB-1::GFP. (A possible titration effect of having additional UAS sequences available for the GAL4.) To control for this possibility of "GAL4 competition", a control experiment should be performed with a UAS-GFP in place of UAS-RREB-1::GFP. The level of expression of the UAS-GFP should be comparable to the UAS-RREB-1::GFP.

We do not believe that *UAS-RREB1::GFP* rescue effect is due to a titration effect of Gal4 by having an additional UAS sequence in this rescue experiment, because *UAS-Oamb,* which also contains an additional UAS sequence, could not rescue the follicle rupture defect (Figure 7), nor did *UAS-hnt^EP55^* (Figure 8—figure supplement 1). In addition, we performed the experiment as reviewers suggested to use a *UAS-GFP* to do the rescue experiment, and it didn’t show any rescue effect as well (Figure 8—figure supplement 3).

Does hnt-overexpression resemble RREB-1 expression? It would be informative to show if there is a phenotype associated with hnt over-expression (using UAS-hnt, UAS-GFP-Hnt, EP55, or P{GSV1}GS1018) in stage 14 follicle cells. Or if not over-expressed per se, is there a consequence to a prematurely high level of hnt expression, or maintenance of a high level of expression, in stage 14C – and could this address whether the down-regulation of hnt expression in 14C stage is functionally significant? I.e. Is hnt over-expression sufficient for the premature rupture phenotype and/or upregulation of Mmp2 and Oamb? Would such a phenotype resemble UAS-RREB1 expression, for example? These are fairly achievable experiments and their omission from this study seems puzzling to me.

We agree with reviewers that these are excellent experiments to carry out. We used *hnt^EP55^* line to perform the rescue and overexpression experiments as *RREB1::GFP* line. Unfortunately, *hnt^RNAi^* lines are strong enough to block any overexpressed *hnt* mRNA through Gal4/UAS system and leave Hnt protein still depleted in rescue experiments, and no rescue effect was observed. The strong depletion of Hnt protein by *hnt^RNAi^* when *hnt^EP55^* was included was further validated using a flip-out actin-Gal4 system. These data are now included in Figure 8—figure supplement 1. In addition, we also observed slight increase of follicle rupture when *hnt* is overexpressed using FC1 Gal4, very similar with overexpression of *RREB1::GFP* with FC1 Gal4. This is consistent with our hypothesis that Hnt functions as competency factor to regulate follicle maturation. The phenotype is weak, which is likely due to the fact that FC1 Gal4 only starts to be expressed in stage 14, when endogenous Hnt already starts expression. It would be ideal to have a Gal4 driver expressed in stage 13 and use such Gal4 driver to overexpress *hnt*, but unfortunately, we are not aware of the existence of this type of Gal4 line. Furthermore, we tried to maintain high *hnt* expression in stage-14C using FC2 Gal4, it caused slight reduction of follicle rupture but no egg-laying defect. Thus, we didn’t make any conclusion to avoid over-interpreting the data. We also performed similar experiments with UAS-Hnt::GFP lines from Dr. Howard Lipshitz’s lab and they showed the same result as *hnt^EP55^* line.

[Editors' note: further revisions were requested prior to acceptance, as described below.]

Reviewer #2:[…] Two questions come to mind:first,is hnt^peb^/+ control (~ 50%) showing a dosage effect compared to + /+ control (~80%)?;

We have repeated this experiment multiple times with two different researchers. Both came out the same result that *hnt^peb^/+; FC2>RFP/+* follicles showed ~50% rupture rate instead of ~80% in wild-type control. Therefore, we think it is possible that *hnt^peb^/+* causes a dosage effect. This dose effect may not just caused by *hnt* in the mature follicle cells, but also from other tissues expressing *hnt* (see response to question 2). Alternatively, *hnt^peb^* may behave as a dominant negative allele since the nature of the temperature sensitivity is unknown.

and second,can the loss-of-function phenotype of hntpeb /hntXE81 and hntpeb /hntEH704a be rescued by FC1 or FC2 GAL4 expression of UAS-RREB-1? This latter experiment would completely and definitively address any concerns regarding effectiveness of RNAi depletion phenotypes and their rescue by UAS-RREB1. Since the rescue of the knockdown is pivotal to a conserved function, I would really like to see inclusion of this experiment. It should only take a couple of fly generations to get the GAL4 + UAS lines into the hntpeb /hntXE81 and hntpeb /hntEH704a backgrounds, and the assay is well established in the lab. I think it will strengthen the rescue results.

We do not believe that FC1 or FC2 driving *UAS-RREB-1* expression will rescue the *hnt* loss-of-function phenotype for the following reasons: Hnt is not only expressed in stage-14 follicle cells, but also expressed in follicle cells of stage 7-stage10B, which is essential for follicle cell differentiation according to my previous study (Sun and Deng, 2007). Although no morphological defect was observed in *hnt^peb/XE81^*and *hnt^peb/EH704a^*mature follicles, we cannot exclude that subtle defect in early development may contribute to the follicle rupture defect. That’s why we used RNAi to bypass earlier Hnt function in our initial experiments. In addition, Hnt is also expressed in the midgut enterocytes and secretory cells in the female reproductive tract. The latter has been demonstrated to regulate ovulation as well (Sun and Spradling, 2013). Hnt may also function in the secretory cells in the female reproductive tract to regulate follicle rupture in addition to its role in the mature follicle cells. This is supported by our recent finding that another transcription factor is functioning in both secretory cells and mature follicle cells to regulate follicle rupture (Knapp and Sun, manuscript in preparation; this story is also going to be presented in the Annual *Drosophila* Research Conference in April 2018). We are interested in investigating the role of Hnt in serectory cells, but this is out of the scope of this study. Therefore, we do not believe this experiment will work. Even this will work, which will take another two months with several generations of crosses, it would not add any new findings to our current conclusion.

Also, is there any evidence that RREB-1 is expressed in the mammalian ovary? Is it possible to find out somehow? (Such as some high throughput data somewhere?) If so, then it would certainly strengthen the conserved function in ovulation. But if it is not? Then…?

I am sure there’re high throughput data available to support that RREB-1 is expressed in mammalian ovary, but many of those data are not conclusive and need further validation. We are interested in this question and our preliminary data had showed that RREB-1 is expressed in follicles in LH-dependent manner. We are in the process of generating RREB-1 conditional knockout mouse to test its role in mammals. This project is out of the scope of this current study.